# MMSI-BENCH: A BENCHMARK FOR MULTI-IMAGE SPATIAL INTELLIGENCE

Sihan Yang[1*]    Runsen Xu[1,2*‡]    Yiman Xie[1,3]    Sizhe Yang[1,2]    Mo Li[1,4]    Jingli Lin[1,5]
Chenming Zhu[1,6]    Xiaochen Chen[7]    Haodong Duan[1]    Xiangyu Yue[1,2]    Dahua Lin[1,2]
Tai Wang[1†]    Jiangmiao Pang[1†]

[1]Shanghai AI Laboratory    [2]The Chinese University of Hong Kong    [3]Zhejiang University    [4]Tsinghua University    [5]Shanghai Jiaotong University    [6]University of Hong Kong    [7]Beijing Normal University
[*] Equal Contribution    [‡] Project Lead    [†] Corresponding Author
taiwang.me@gmail.com    pangjiangmiao@gmail.com

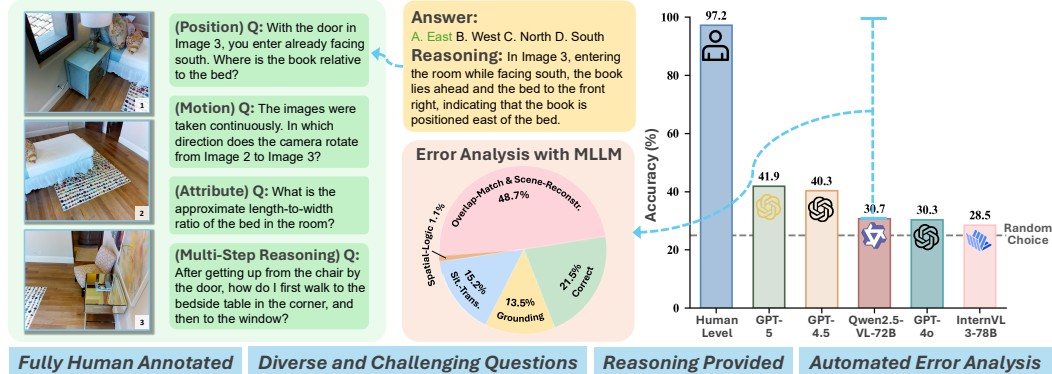

Figure 1: **The MMSI-Bench benchmark. Left and middle:** Human experts carefully select image sets to create unambiguous and challenging multi-image spatial reasoning questions. Human-annotated reasoning processes are also provided for correctness verification and to further facilitate automated error analysis. **Right:** Comparison of models and human performance on MMSI-Bench, highlighting the benchmark's difficulty and the substantial gap between current MLLMs and human-level spatial intelligence.

## ABSTRACT

Spatial intelligence is essential for multimodal large language models (MLLMs) operating in the complex physical world. Existing benchmarks, however, probe only single-image relations and thus fail to assess the multi-image spatial reasoning that real-world deployments demand. We introduce MMSI-Bench, a VQA benchmark dedicated to multi-image spatial intelligence. Six 3D-vision researchers spent more than 300 hours meticulously crafting 1,000 challenging, unambiguous multiple-choice questions from over 120,000 images, each paired with carefully designed distractors and a stepwise reasoning process. We conduct extensive experiments and evaluate 37 open-source and proprietary MLLMs, observing a wide gap: the strongest open-source model attains roughly 30% accuracy and OpenAI's GPT-5 reasoning model reaches 40%, while humans score 97%. These results underscore the challenging nature of MMSI-Bench and the substantial headroom for future research. Leveraging the annotated reasoning processes, we also provide an automated error analysis pipeline that diagnoses four dominant failure modes, including (1) grounding errors, (2) overlap-matching and scene-reconstruction errors, (3) situation-transformation reasoning errors, and (4) spatial-logic errors, offering insights for advancing spatial intelligence.

# 1 INTRODUCTION

Multimodal large language models (MLLMs) have rapidly advanced their ability to ground language and vision in the physical world (Liu et al., 2024a; Black et al., 2024; Peng et al., 2023), making them a promising stepping-stone toward embodied artificial general intelligence (AGI). A core capability of MLLMs deployed in the physical world is spatial intelligence, the skill to understand where things are, how they move, etc. To steer progress toward this goal, the community needs rigorous, diverse, and challenging evaluation benchmarks that can serve as north stars for progress.

However, most existing benchmarks probe only single-image spatial understanding and restrict themselves to straightforward spatial relations (Liu et al., 2023; Tong et al., 2024; Johnson et al., 2017; Cheng et al., 2024; Chen et al., 2024a). Real-world spatial understanding is more complex: models must reason across multiple images to track object and ego motion, and relate entities that never co-occur in a single frame (Xu et al., 2024; Wang et al., 2024b; Lyu et al., 2024). Without benchmarks that demand such multi-image reasoning, we cannot reliably measure or improve an MLLM's spatial competence.

In this work, we introduce MMSI-Bench, a multiple-choice VQA benchmark dedicated to **M**ulti-i**M**age-based **S**patial **I**ntelligence (Figure 1). MMSI-Bench is built on a comprehensive taxonomy of ten fundamental spatial reasoning tasks that span the positions, attributes, and motions of three key spatial elements—camera (observer), object, and region in real-world scenes. Beyond these atomic tasks, we construct a multi-step reasoning split that chains them into long-horizon questions.

Crafting a benchmark that is simultaneously diverse, accurate, and challenging is far from trivial. Prior benchmarks (Chen et al., 2024a; Cheng et al., 2024; Yang et al., 2024; Fu et al., 2024) populate templated questions from annotated or estimated spatial metadata. These automatic pipelines suffer from limited diversity and difficulty. We therefore adopt a fully human-centric design. For each question, one of six 3D-vision researchers scours large image collections, selects a set of images that exhibit non-trivial spatial relations, and creates a novel, difficult, and unambiguous multiple-choice question whose answer can be derived only by reasoning across all selected images. Every item comes with carefully designed distractors and a step-by-step reasoning process that explicitly leads to the correct answer, a second annotator then audits each example to ensure clarity and correctness.

Over the course of 300+ researcher-hours, we inspected more than 120,000 candidate images and distilled them into 1,000 high-quality Q&A pairs covering a broad spectrum of real-world scenes, from indoor scans (Dai et al., 2017; Chang et al., 2017) and outdoor driving footage (Sun et al., 2020; Caesar et al., 2020) to robot manipulation (AgiBot-World-Contributors et al., 2025) and everyday activities (Grauman et al., 2022). We evaluate 37 widely used MLLMs on MMSI-Bench. The strongest open-source model attains about 30% accuracy, while the best proprietary system, the OpenAI GPT-5 reasoning model (OpenAI, 2025c), reaches only 40%, compared with human performance at 97% accuracy. To the best of our knowledge, MMSI-Bench shows the largest performance margin between SOTA models and humans among existing spatial intelligence benchmarks.

This pronounced gap underscores both the difficulty of MMSI-Bench and the headroom for progress in spatial intelligence. Our error analysis groups failures into four categories: (1) grounding errors, (2) overlap-matching and scene-reconstruction errors, (3) situation-transformation reasoning errors, and (4) spatial-logic errors. Because every question is paired with a reference reasoning process, we also release a novel automated analysis pipeline that contrasts a model's response with the reference to pinpoint error types efficiently across the entire benchmark, providing actionable insights for future work on multi-image spatial intelligence.

# 2 RELATED WORK

**Multi-image VQA benchmarks.** Driven by the rapid progress of MLLMs, VQA benchmarks have shifted from single-image reasoning (Hudson & Manning, 2019) to multi-image settings focusing on various skills. Visual Haystacks (Wu et al., 2025b) probes long-context retrieval, asking models to locate relevant content within large image collections. MIBench (Liu et al., 2024b) stresses cross-image reasoning that combines instruction following with knowledge retrieval. ReMI (Kazemi et al., 2024) broadens the scope to math, physics, and code across two to six images, blending GPT-generated items with human audits. BLINK (Fu et al., 2024) zeroes in on inter-image relationships, while

MuirBench (Wang et al., 2024a) adds robustness and diversity to multi-image reasoning. Existing benchmarks show that contemporary MLLMs still trail behind humans as either the number of images or the reasoning complexity grows (Li et al., 2023). In this work, instead of building a general benchmark, we focus on comprehensively evaluating MLLMs' multi-image spatial intelligence.

**Benchmarks for spatial intelligence of MLLMs.** Recent progress in MLLMs has enabled them to potentially serve as the "brains" of embodied agents, e.g., the decision-making modules of autonomous vehicles (Sima et al., 2024) or robots (Kim et al., 2024; Team et al., 2025a; Black et al., 2024), where reliable spatial intelligence is indispensable. To measure this capability, a variety of benchmarks have been proposed, but the majority are designed to assess only single-image spatial understanding (Liao et al., 2024; Liu et al., 2023; Shiri et al., 2024; Zhou et al., 2025; Ma et al., 2024; Song et al., 2024; Du et al., 2024; Tong et al., 2024; Zhang et al., 2024). Beyond a single view, associating information across multiple images is even more critical in practice, yet existing multi-image benchmarks remain fragmentary. BLINK (Fu et al., 2024) and MuriBench (Wang et al., 2024a) contain just a few spatial sub-splits within otherwise general VQA suites, and thus lack comprehensive coverage. LEGO-Puzzles (Tang et al., 2025) probes multi-step reasoning in a synthetic LEGO-assembly setting, overlooking real-world scenes. Efforts such as UniQA-3D (Zuo et al., 2024), MMIU (Meng et al., 2024), SAT (Ray et al., 2024), MultiSPA (Xu et al., 2025), and VSI-Bench (Yang et al., 2024) generate template-based or rule-based questions from existing annotations/metadata or simulators, which constrains question diversity and difficulty. By contrast, MMSI-Bench is fully human-curated, unconstrained by templates, and features challenging, diverse questions with accompanying reasoning annotations. ERQA (Team et al., 2025b) is likewise human-curated but comprises only 400 samples, of which just 113 are multi-image. Detailed quantitative comparisons (e.g., diversity) are provided in Appendix D.

## 3 MMSI-BENCH

In this section, we present MMSI-Bench, a comprehensive and challenging benchmark designed to evaluate the multi-image spatial reasoning capability of MLLMs. Specifically, we introduce the concept of multi-image spatial reasoning and outline its associated sub-tasks in Section 3.1. In Section 3.2, we provide a detailed description of the benchmark construction process.

### 3.1 OVERVIEW OF MMSI-BENCH

Table 1: Taxonomy of Multi-Image Spatial Reasoning.

| Main Category | Sub-Category | Description |
|---|---|---|
| Positional Relationship | Camera-Camera | Relationship between two camera viewpoints. |
| | Camera-Object | Relationship between the camera and an object. |
| | Camera-Region | Relationship between the camera and a semantic region (e.g., kitchen). |
| | Object-Object | Relationship between two objects. |
| | Object-Region | Relationship between an object and a semantic region. |
| | Region-Region | Relationship between two different regions. |
| Attribute | Measurement | Geometric properties (e.g., length, size) of elements. |
| | Appearance | Visual characteristics (e.g., shape) of elements. |
| Motion | Camera | The camera's movement (e.g., direction). |
| | Object | The movement of objects within the scene. |
| Multi-Step Reasoning | — | Complex tasks requiring a sequence of reasoning steps (e.g. navigation). |

Humans possess the remarkable ability to infer spatial information about their environment by integrating visual inputs from different viewpoints or moments in time. We refer to this process as multi-image spatial reasoning, in which information from multiple images is combined to reconstruct the underlying scene and enable reasoning to answer complex spatial questions. To evaluate the multi-image spatial reasoning capability of multimodal large language models, we introduce MMSI-Bench. We further discuss additional motivations for multi-image input in Appendix C.

We categorize tasks based on three fundamental spatial elements: **camera** (the agent that captures the scenes), **object** (entities other than the agent within the scenes), and **region** (semantic areas within the scenes, such as a kitchen). Fundamental spatial understanding requires not only recognizing the **positional relationships** among these elements but also their **attributes** and potential **motion**. Guided by this taxonomy, we define ten spatial reasoning task types and one multi-step spatial reasoning

category, which flexibly combines these basic types. The specific categories are presented in Table 1. Examples for all task categories are provided in Figure 2. Notably, our questions are designed to be human-answerable; thus, we do not include questions about camera attributes (parameters). Additionally, since regions are generally static, we do not include motion-related tasks for regions.

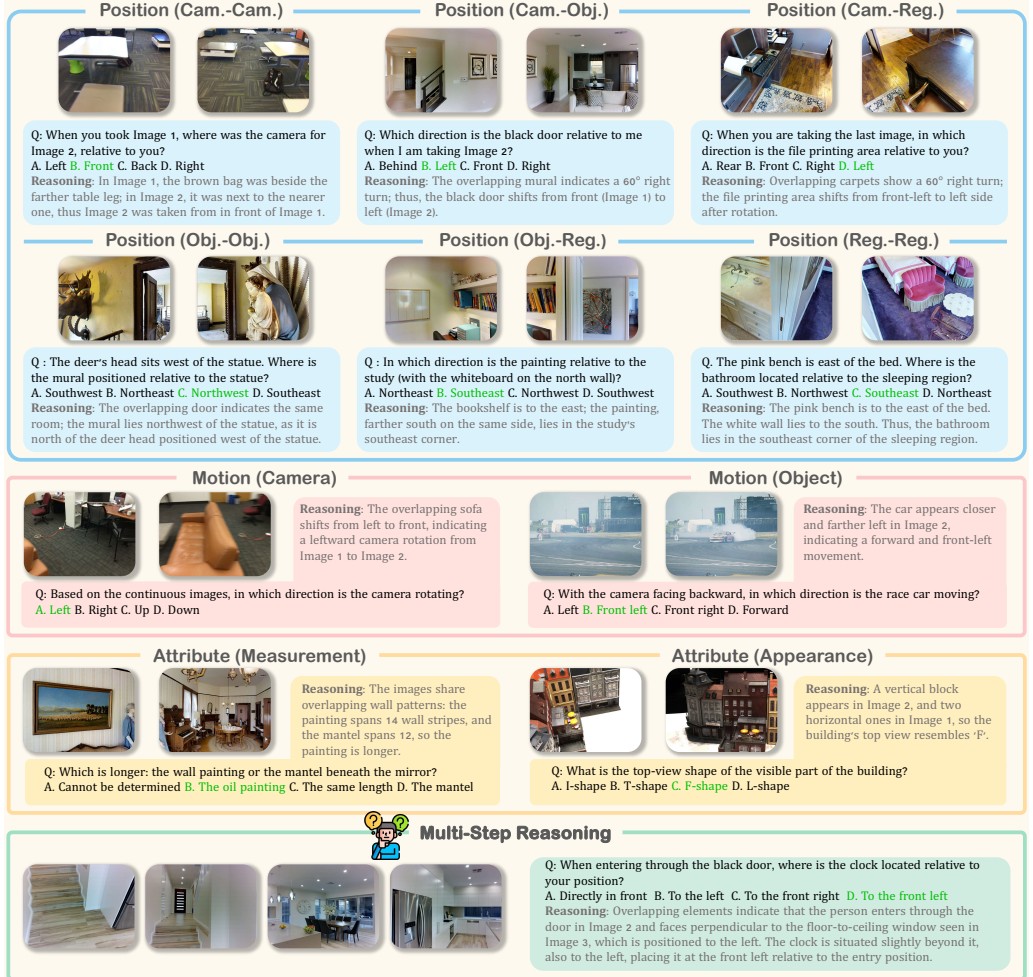

Figure 2: Representative MMSI-Bench samples from each category. Please zoom in to inspect image details. Questions and rationales are simplified for brevity; the complete versions appear in Appendix G. Correct answers are highlighted in green.

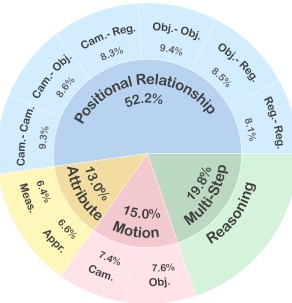

Figure 3: Distribution of categories in MMSI-Bench.

Table 2: Dataset statistics.

| Statistic | Value |
|---|---|
| Total questions | 1,000 |
| Number of unique images | 1,990 |
| Average question length | 130.28 |
| Average reasoning process length | 252.83 |
| Average image count per question | 2.55 |
| Maximum question length | 449 |
| Maximum reasoning process length | 1,118 |
| Maximum image count per question | 10 |

**Data sources:** ScanNet, nuScenes, Matterport3D, Ego4D, AgiBot-World, DTU, DAVIS 2017, Waymo.

In conclusion, MMSI-Bench consists of 1,000 multiple-choice question–answer pairs based on multiple images, each also accompanied by annotated reasoning processes explaining the answer.

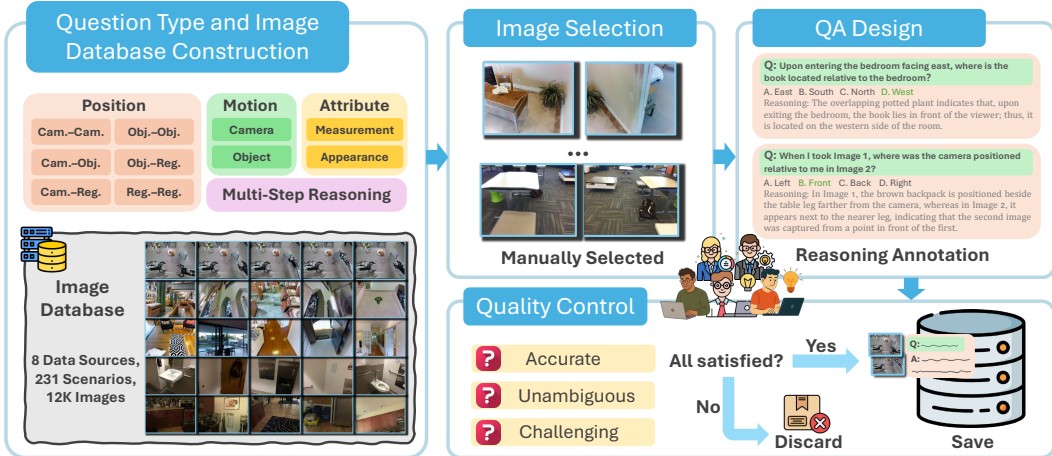

Figure 4: Illustration of the MMSI-Bench construction pipeline: images are collected from diverse real-world datasets, relevant image sets are carefully selected, complex QA tasks and detailed reasoning processes are manually annotated, and all data undergo rigorous quality control.

Every question is specifically designed such that it cannot be answered using a single image alone. Except for the Multi-Step Reasoning category, all other categories contain only two images per question. Detailed statistics on the proportion of questions in each category and other relevant information about MMSI-Bench are presented in Figure 3 and Table 2, respectively.

## 3.2 BENCHMARK CONSTRUCTION PROCESS

Our detailed benchmark construction process is illustrated in Figure 4.

**Data collection.** To comprehensively evaluate the capability of MLLMs across a variety of real-world scenarios, it is essential to curate data from diverse and representative sources. To ensure the complexity and authenticity of the evaluation, we exclusively utilize images captured from real-world scenes. Accordingly, we leverage images from a diverse collection of real-world datasets: indoor 3D scene datasets (Matterport3D (Chang et al., 2017) and ScanNet (Dai et al., 2017)), a local scene reconstruction dataset (DTU (Jensen et al., 2014)), autonomous driving datasets (nuScenes (Caesar et al., 2020), Waymo (Sun et al., 2020)), a robotics dataset (AgiBot-World (AgiBot-World-Contributors et al., 2025)), a video object segmentation dataset (DAVIS 2017 (Pont-Tuset et al., 2017)), and an egocentric video dataset (Ego4D (Grauman et al., 2022)). These diverse data sources enable the construction of various question-answer pairs, ensuring that our benchmark captures the complexity and diversity inherent in real-world environments.

**Question-answer design and reasoning process annotation.** Using the collected datasets, six 3D vision researchers annotate QA pairs spanning the above eleven distinct task categories. Annotators are explicitly instructed to create a diverse set of questions that are as novel and as challenging as possible for MLLMs, aiming to push the boundaries of current model capabilities. At the same time, annotators are instructed to ensure that all questions are accurate, unambiguous, and answerable by humans. For each question, annotators begin by visually inspecting the image database to choose a set of relevant images. They then construct a question that cannot be answered based on any single image in isolation; rather, the question requires integrating information from multiple images in order to infer the actual scenario. To ensure diversity in question formats, no templates are used, and all questions are created in free form. Each question is designed as a four-option multiple-choice problem, with only one correct answer. The incorrect options are also carefully crafted to serve as plausible distractors. In addition, annotators provide detailed reasoning processes alongside each answer, which are used to facilitate filtering incorrect samples during quality control and to enable automated analysis of MLLMs' reasoning processes, as described in Section 5.2.

**Quality control.** To ensure the quality of the data, three additional reviewers, independent of the original annotators, systematically examine all data and remove any instances containing ambiguous questions (due to linguistic ambiguities or insufficient visual information), incorrect answers, or questions that can be answered using a single image or common sense alone. This rigorous quality

control ensures our benchmark provides a reliable, high-quality evaluation. We annotate the difficulty level of the data based on human answering time; detailed information is provided in Appendix F.6.

# 4   EVALUATION ON MMSI

## 4.1   EVALUATION SETUP

We evaluate 37 open-source and proprietary MLLMs on MMSI-Bench. For consistency, all models are evaluated with a temperature of zero and a maximum output length of 2,048 tokens.

For comparison, we include three baselines: **Random Guessing**, **Human Performance**, and **Blind GPT-4o**. Random Guessing reflects the expected accuracy under equal-probability selection from all options. Human performance is measured as the average accuracy of five independent evaluators not involved in data annotation, representing human-level spatial intelligence on this benchmark. Blind GPT-4o denotes the performance of GPT-4o without access to image inputs.

We report accuracy (%) using exact match between answers extracted from model outputs and ground-truth answers for our multiple-choice questions. If a model fails to generate an answer in the required format, we adopt the LLM-based fallback strategy from VLMEvalKit (Duan et al., 2024) to extract the intended response. More experimental details are provided in Appendix E.

## 4.2   MAIN RESULTS

Table 3 presents the primary results of our evaluation. We summarize our main findings as follows:

**Current MLLMs struggle with multi-image spatial reasoning.** Despite recent progress, even the most advanced MLLMs exhibit significant limitations in multi-image spatial reasoning. Most models achieve relatively low average scores, often only marginally outperforming random guessing, and are far from human-level performance. For example, the best-performing proprietary reasoning model, GPT-5, achieves an average accuracy of only 41.9%, while human performance exceeds 97%, highlighting a substantial gap in spatial intelligence between current MLLMs and humans. Additionally, the accuracy of Blind GPT-4o is close to that of random guessing, indicating that our tasks indeed require genuine visual-spatial reasoning and cannot be solved by language priors or commonsense knowledge alone.

**Advanced open-source models still trail proprietary counterparts.** We observe that leading open-source models generally underperform compared to proprietary models. For instance, the best open-source model, Qwen2.5-VL-72B, achieves an average score of 30.7%, which is notably lower than that of the top proprietary model. This underscores a notable performance gap between cutting-edge open-source and proprietary MLLMs on this challenging benchmark.

**Multi-step reasoning and camera motion comprehension are challenging.** For most models, performance on multi-step reasoning (MSR) tasks is lower than on single-step tasks (i.e., the average performance on Positional Relationship, Attribute, and Motion categories), indicating that integrating information across multiple reasoning steps remains a challenge for current MLLMs. In addition, most models, particularly open-source ones, perform poorly on camera motion tasks, suggesting that MLLMs, as embodied agents, struggle to understand their own movement. One potential reason is that open-source models generally lack access to annotated first-person motion data during training, while proprietary models may leverage more diverse datasets containing such information.

**Scaling up model size brings limited gains on MMSI-Bench.** We observe that, for models within the same series, increasing the number of parameters leads to only limited performance gains on MMSI-Bench. For example, Qwen2.5-VL-72B achieves only a 3% higher accuracy than Qwen2.5-VL-32B, and InternVL3-78B outperforms InternVL3-1B by just 1.5%. Moreover, NVILA-15B surpasses most models with over 70 billion parameters, ranking second only to the best-performing open-source model, Qwen2.5-VL-72B. These results suggest that, at present, the bottleneck for advancing MLLM performance on complex spatial reasoning tasks may lie primarily in data quality and diversity, rather than model size. How to best leverage data, model size, and architecture in synergy remains an open and important question for future research.

Table 3: Evaluation results for 37 MLLMs on MMSI-Bench. For each category, the best-performing proprietary model and the best-performing open-source model are both indicated in **bold**.

| Models | Positional Relationship | | | | | | Attribute | | Motion | | MSR | Avg. |
| --- | --- | --- | --- | --- | --- | --- | --- | --- | --- | --- | --- | --- |
| | Cam.–Cam. | Obj.–Obj. | Reg.–Reg. | Cam.–Obj. | Obj.–Reg. | Cam.–Reg. | Meas. | Appr. | Cam. | Obj. | – | |
| *Proprietary* | | | | | | | | | | | | |
| GPT-5 | 43.0 | 35.1 | 32.1 | 48.8 | 42.4 | 51.8 | 60.9 | **36.4** | 32.4 | 36.8 | **42.0** | 41.9 |
| o3 | **45.2** | **39.4** | 37.0 | 44.2 | 47.1 | **62.6** | 54.7 | 28.8 | 31.1 | 32.9 | 34.9 | 41.0 |
| GPT-4.5 | 34.4 | 29.8 | **39.5** | **51.2** | 47.1 | 55.4 | 39.1 | 33.3 | **41.9** | **40.8** | 36.4 | 40.3 |
| GPT-4.1 | 36.6 | 26.6 | 27.2 | 29.1 | 36.5 | 27.7 | 37.5 | 24.2 | 36.5 | 32.9 | 28.8 | 30.9 |
| GPT-4o | 34.4 | 24.5 | 23.5 | 19.8 | 37.6 | 27.7 | 32.8 | 31.8 | 35.1 | 36.8 | 30.8 | 30.3 |
| Gemini-2.5-Pro | 39.7 | 31.9 | **39.5** | 45.3 | 35.2 | 43.3 | 51.5 | 21.2 | 36.4 | 30.2 | 34.3 | 36.9 |
| Claude-3.7-Sonnet | 35.5 | 29.8 | 29.6 | 31.4 | 32.9 | 34.9 | 37.5 | 25.8 | 23.0 | 31.6 | 25.8 | 30.2 |
| dots.vlm 1 | 33.3 | 31.9 | 38.3 | 27.9 | 34.1 | 42.2 | **62.5** | **36.4** | 23.0 | 25.0 | 30.8 | 34.1 |
| GLM-4.5V-thinking | 38.7 | 35.1 | 29.6 | 30.2 | 40.0 | 39.8 | 45.3 | 28.8 | 25.7 | 32.9 | 30.3 | 33.8 |
| Doubao-1.5-pro | 25.8 | 33.0 | 33.3 | 33.7 | **49.4** | 39.8 | 37.5 | 31.8 | 27.0 | 26.3 | 29.8 | 33.0 |
| Seed1.5-VL | 32.2 | 30.8 | 25.9 | 23.2 | 38.8 | 32.5 | 39.0 | 21.2 | 36.4 | 25.0 | 26.2 | 29.7 |
| *Open-source* | | | | | | | | | | | | |
| InternVL3-78B | 34.4 | 23.4 | 32.1 | 12.8 | 37.6 | 26.5 | 37.5 | 19.7 | **28.4** | 31.6 | 29.3 | 28.5 |
| InternVL2.5-78B | 23.7 | 22.3 | **39.5** | 29.1 | 31.8 | **42.2** | 35.9 | 19.7 | 17.6 | 26.3 | 27.3 | 28.5 |
| Qwen2.5-VL-72B | 25.8 | **34.0** | 34.6 | 23.3 | 34.1 | 36.1 | **45.3** | 27.3 | 27.0 | 30.3 | 27.3 | **30.7** |
| LLaVA-OneVision-72B | **43.0** | 31.9 | 33.3 | 30.2 | 37.6 | 38.6 | 28.1 | 19.7 | 13.5 | 32.9 | 15.7 | 28.4 |
| InternVL3-38B | 21.5 | 20.2 | 33.3 | 23.3 | 35.3 | 25.3 | 39.1 | 21.2 | 16.2 | 31.6 | 25.8 | 26.3 |
| InternVL2.5-38B | 18.3 | 22.3 | 35.8 | 22.1 | **38.8** | 34.9 | 37.5 | 25.8 | 14.9 | 38.2 | 25.3 | 27.9 |
| Qwen2.5-VL-32B | 24.7 | 26.6 | 29.6 | 22.1 | 32.9 | 31.3 | 31.2 | 24.2 | 18.9 | 35.5 | 27.8 | 27.7 |
| InternVL2.5-26B | 24.7 | 19.1 | 29.6 | 33.7 | 31.8 | 37.3 | 35.9 | 30.3 | 10.8 | 31.6 | 26.8 | 28.0 |
| NVILA-15B | 30.1 | 39.4 | 28.4 | 36.0 | **38.8** | 20.5 | 29.7 | **31.8** | 18.9 | 35.5 | 27.8 | 30.5 |
| InternVL3-14B | 19.4 | 24.5 | 24.7 | 23.3 | 37.6 | 24.1 | 31.2 | 22.7 | 24.3 | 31.6 | 29.3 | 26.8 |
| Llama-3.2-11B-Vision | 25.8 | 30.8 | 32.0 | 25.6 | 21.2 | 25.9 | 20.3 | 19.7 | 25.6 | 28.9 | 19.2 | 25.4 |
| InternVL3-9B | 18.3 | 25.5 | 32.1 | 29.1 | 31.8 | 22.9 | 29.7 | 24.2 | 16.2 | 38.2 | 26.8 | 26.7 |
| InternVL3-8B | 25.8 | 31.9 | 37.0 | 25.6 | 35.3 | 28.9 | 23.4 | 24.2 | 16.2 | 32.9 | 14.6 | 25.7 |
| InternVL2.5-8B | 32.3 | 27.7 | 29.6 | 32.6 | 24.7 | 32.5 | 26.6 | 27.3 | 16.2 | 31.6 | **30.3** | 28.7 |
| NVILA-8B | 17.2 | 29.8 | 24.7 | 30.2 | 22.4 | 34.9 | 34.4 | 25.8 | 25.7 | 34.2 | 29.8 | 28.1 |
| Qwen2.5-VL-7B | 24.7 | 24.5 | 24.7 | 25.6 | 29.4 | 26.5 | 25.0 | 18.2 | 20.3 | **39.5** | 25.8 | 25.9 |
| LLaVA-OneVision-7B | 20.4 | 33.0 | 26.6 | 29.1 | 25.9 | 30.1 | 29.7 | 25.8 | 18.9 | 34.2 | 11.6 | 24.5 |
| InternVL2.5-4B | 31.2 | 23.4 | 21.0 | 31.4 | 34.1 | 25.3 | 23.4 | 24.2 | 13.5 | 31.6 | 26.8 | 26.3 |
| Qwen2.5-VL-3B | 26.9 | 27.7 | 30.9 | 29.1 | 28.2 | 34.9 | 31.2 | 16.7 | 17.6 | 27.6 | 23.2 | 26.5 |
| InternVL3-2B | 26.9 | 25.5 | 29.6 | 31.4 | 28.2 | 27.7 | 26.6 | 22.7 | 12.2 | 23.7 | 23.7 | 25.3 |
| InternVL2.5-2B | 28.0 | 27.7 | 24.7 | **37.2** | 29.4 | 36.1 | 43.8 | 15.2 | 21.6 | 31.6 | 26.8 | 29.0 |
| InternVL3-1B | 24.7 | 35.1 | 22.2 | 30.2 | 29.4 | 30.1 | 32.8 | 28.8 | 17.6 | 19.7 | 26.3 | 27.0 |
| InternVL2.5-1B | 23.7 | 26.6 | 24.7 | 25.6 | 31.8 | 25.3 | 31.2 | 30.3 | 17.6 | 25.0 | 26.3 | 26.1 |
| DeepSeek-VL2 | 23.7 | 31.9 | 22.2 | 36.0 | 30.6 | 22.9 | 28.1 | 15.2 | **28.4** | 26.3 | 28.3 | 27.1 |
| DeepSeek-VL2-Small | 24.7 | 28.7 | 18.5 | 33.7 | **38.8** | 27.7 | 28.1 | 33.3 | 24.3 | 25.0 | 29.8 | 28.6 |
| DeepSeek-VL2-Tiny | 29.0 | 27.7 | 21.0 | 23.3 | 17.6 | 31.3 | 14.1 | 24.2 | 14.9 | 25.0 | 27.3 | 24.0 |
| *Baseline* | | | | | | | | | | | | |
| Blind GPT-4o | 20.2 | 17.0 | 29.6 | 13.9 | 29.4 | 19.2 | 21.8 | 12.1 | 20.2 | 29.0 | 20.2 | 22.7 |
| Random Guessing | 25.0 | 25.0 | 25.0 | 25.0 | 25.0 | 25.0 | 25.0 | 25.0 | 25.0 | 25.0 | 25.0 | 25.0 |
| Human Level | 95.7 | 98.9 | 97.5 | 94.2 | 98.8 | 96.4 | 95.3 | 98.5 | 98.6 | 98.7 | 97.0 | 97.2 |

Table 4: Performance of models fine-tuned on spatial-reasoning data and their base models on MMSI-Bench.

| Models | Positional Relationship | | | | | | Attribute | | Motion | | MSR | Avg. |
| --- | --- | --- | --- | --- | --- | --- | --- | --- | --- | --- | --- | --- |
| | Cam.–Cam. | Obj.–Obj. | Reg.–Reg. | Cam.–Obj. | Obj.–Reg. | Cam.–Reg. | Meas. | Appr. | Cam. | Obj. | – | |
| Qwen2.5-VL 3B | 26.9 | 27.7 | 30.9 | 29.1 | 28.2 | 34.9 | 31.2 | 16.7 | 17.6 | 27.6 | 23.2 | 26.5 |
| Spatial-MLLM 3B (Wu et al., 2025a) | 23.6 | 26.6 | 37.0 | 34.9 | 32.9 | 24.1 | 18.8 | 34.9 | 34.9 | 25.0 | 28.8 | 27.7 |
| InternVL2_5-8B | 32.3 | 27.7 | 29.6 | 32.6 | 24.7 | 32.5 | 26.6 | 27.3 | 16.2 | 31.6 | 30.3 | 28.7 |
| InternSpatial (Deng et al., 2025) | 20.4 | 27.6 | 28.4 | 29.0 | 30.5 | 26.5 | 29.6 | 27.2 | 14.8 | 23.6 | 31.3 | 26.9 |
| Qwen2.5-VL-32B | 24.7 | 26.6 | 29.6 | 22.1 | 32.9 | 31.3 | 31.2 | 24.2 | 18.9 | 35.5 | 27.8 | 27.7 |
| RoboBrain2.0 32B (Team et al., 2025a) | 21.5 | 28.7 | 27.1 | 22.0 | 27.0 | 32.5 | 31.2 | 36.3 | 36.4 | 23.6 | 29.2 | 28.5 |

**Current spatial fine-tuning yields limited benefits on MMSI-Bench.** We evaluated models fine-tuned on spatial-reasoning data on MMSI-Bench (detailed results are provided in Table 4) and found only marginal gains over their base counterparts, underscoring the benchmark's high difficulty and stringent demands for spatial reasoning generalization. A likely reason is that their training data, built from limited templates, fails to generalize to the diverse and challenging scenarios in MMSI-Bench.

## 4.3 LIMITED EFFECT OF PROMPTING TECHNIQUES ON MMSI-BENCH

Prompting techniques are widely adopted to improve the reasoning and problem-solving capabilities of large models. Building on these advances, we investigate whether such methods can similarly

enhance the spatial reasoning abilities of MLLMs on MMSI-Bench. Specifically, we investigate both linguistic and visual prompting approaches (see Appendix E.2 for further details):

**Linguistic prompting:** We adopt the widely used Zero-Shot Chain-of-Thought (CoT) method. Following prior work (Wei et al., 2022), we prepend "Let's think step by step" to the prompts to encourage reasoning.

**Visual prompting:** For multi-image spatial understanding, a fundamental requirement is the ability to identify overlaps between images in order to establish relationships across them. However, MLLMs may struggle to identify overlaps between images. To explicitly provide this cue to the model, we employ the popular image matching method PATS (Ni et al., 2023) to establish sparse correspondences between pairs of images, which involves concatenating multiple images and drawing lines between corresponding points to explicitly indicate cross-image relationships. As a baseline, we use concatenated images without correspondence lines as visual inputs, and compare their results with those obtained using visual prompt.

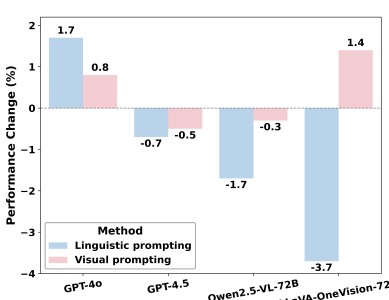

Figure 5: Impact of linguistic and visual prompting on MMSI-Bench.

We conduct experiments using 4 representative models. As shown in Figure 5, the linguistic prompting approach (CoT) leads to a modest performance improvement only for GPT-4o, while resulting in performance degradation on the other models. This limited effectiveness of linguistic prompting may be attributed to the models' inherent lack of spatial reasoning capabilities; simply encouraging "thinking" does not address their fundamental limitations in spatial understanding.

Similarly, the visual prompting approach yields only slight performance improvements for two models, while resulting in performance degradation for the other two. One possible explanation is that most models still lack the basic spatial intelligence required to benefit from such cues: they may not recognize the need to identify overlaps between images to establish spatial relationships, or, even when overlaps are identified, they struggle to reconstruct the actual scene. These findings indicate that further progress in the spatial reasoning ability of MLLMs may depend on advances in model architectures or training paradigms, such as the incorporation of domain-specific data, rather than relying solely on improved prompting strategies.

## 5 ERROR ANALYSIS

This section analyzes the errors made by MLLMs on MMSI-Bench. First, Section 5.1 presents a manual categorization of error types observed in the reasoning processes of the representative model. Then, in Section 5.2, we introduce an enhanced automated error analysis method for systematically examining the reasoning errors of various MLLMs with the help of our annotated reasoning processes.

### 5.1 ERROR TYPE CATEGORIZATION

To identify the main bottlenecks of MLLMs on our benchmark, we manually analyze the reasoning processes exhibited by a representative model, GPT-4o, when prompted with chain-of-thought (CoT) instructions. Specifically, we randomly sample 100 instances from MMSI-Bench and categorize the observed errors into four distinct types (as illustrated in Figure 6):

**Grounding errors.** The model fails to correctly identify or localize relevant objects or details within the image. For instance, it might misclassify an object or incorrectly determine the position of an object, errors that undermine its ability to anchor visual reasoning in the actual image content.

**Overlap-matching and scene-reconstruction errors.** The model fails to identify and match corresponding points that represent the same locations or objects in the real scene across different images, and struggles to implicitly reconstruct the underlying scene based on these cross-image relationships. For example, it cannot recognize that a tree appearing in both images is actually the same tree, leading to an incorrect reconstruction of the scene layout.

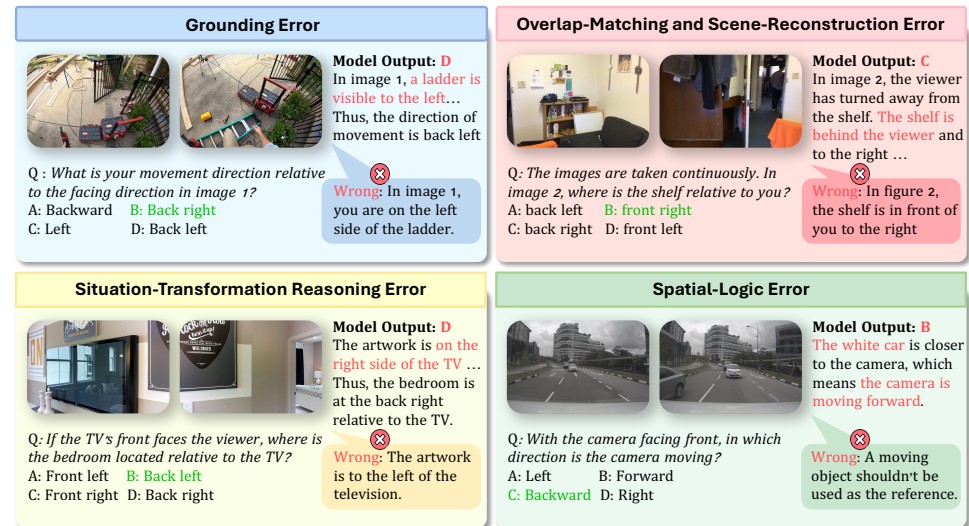

Figure 6: Illustration of four error types identified in MLLM spatial reasoning on MMSI-Bench.

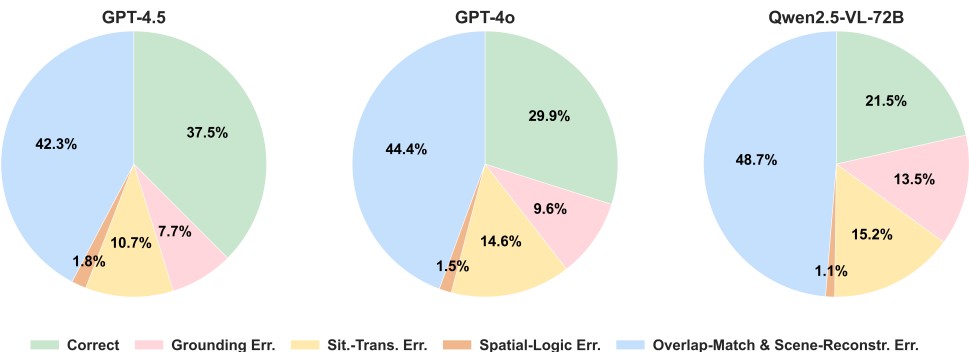

Figure 7: Distribution of correct and error types across three representative MLLMs.

**Situation-transformation reasoning errors.** The model makes mistakes when reasoning about spatial directions relative to different reference objects, or when converting between relative directions (such as left and right) and absolute directions (such as north, south, east, and west).

**Spatial-logic errors.** The model exhibits errors in spatial logical reasoning, such as (1) hallucinating spatial relationships that do not exist in the given context; (2) failing to correctly apply the transitivity of spatial relations, for example, given that A is east of B and B is east of C, the model incorrectly infers that A is west of C; and (3) making mistakes in identifying the correct reference object during motion reasoning, such as using another moving object as the reference point when determining an object's absolute motion status (whereas the correct approach would be to use a stationary object like the ground as the reference).

## 5.2 AUTOMATED ERROR ANALYSIS

Error analysis of model outputs can provide valuable insights for improving MLLMs' spatial capabilities. However, manually reviewing each output is time-consuming and impractical at scale.

We try to give the model the correct answer and ask it to reflect on its output and identify the primary error type following Section 5.1. In experiments with GPT-4o, it correctly labels the error type for only 53% of the selected samples, indicating that the correct answer alone is insufficient for evaluating the reasoning process. When we also supply the human-annotated reasoning process, the accuracy rises to 78%, highlighting the effectiveness and essential role of our reasoning annotation in achieving reliable automated error analysis.

We then apply this approach to analyze the reasoning processes of representative models across all MMSI-Bench questions, with results presented in Figure 7, which reveals the distribution of error

types across different models and highlights specific directions for improving their spatial reasoning abilities. Notably, we observe that even when the final answer is correct, the reasoning process can still contain significant errors. Specifically, the reasoning accuracy of GPT-4.5, GPT-4o, and Qwen2.5-VL-72B is 37.5%, 29.9%, and 21.5%, respectively, all lower than their corresponding multiple-choice accuracy. In particular, Qwen2.5-VL-72B's reasoning accuracy is about 10% lower than its multiple-choice accuracy, revealing its insufficient spatial reasoning capabilities. Among the various error types, overlap-matching and scene-reconstruction errors account for the largest proportion of mistakes across all models. This suggests a clear direction for future progress as MLLMs strive for human-level spatial intelligence.

## 6 CONCLUSION

To summarize, we introduce MMSI-Bench, a challenging and comprehensive benchmark specifically designed to assess the multi-image spatial reasoning capability of MLLMs. Our evaluation across 37 state-of-the-art models reveals a substantial gap between current MLLMs and human-level spatial intelligence. Leveraging expert-annotated reasoning processes, we further propose an automated error analysis pipeline that enables scalable and systematic diagnosis of model failures, highlighting concrete directions for improving spatial reasoning in future models. We hope MMSI-Bench will serve as a valuable resource for the community and accelerate progress toward more spatially capable and robust multimodal AI systems. We further discuss the LLM usage and limitations in Appendix B.

ACKNOWLEDGMENTS

This work is funded in part by the National Key R&D Program of China, and Shanghai Artificial Intelligence Laboratory.

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

## A    APPENDIX OVERVIEW AND ORGANIZATION

This appendix provides supplementary details to support and extend the main paper. The organization of the appendix is as follows:

1. **LLM Usage, Limitations, Social Impact, and License & Access (Section B):** This section outlines not only our LLM Usage, the limitations of our work, but also discusses its potential social implications, associated risks, and ethical considerations. In addition, it clarifies the licensing terms and access conditions for MMSI-Bench, ensuring responsible and transparent use of the dataset.

2. **Motivation for Multi-Image Input (Section C):** We discuss the unique advantages of multi-image input for evaluating spatial intelligence.

3. **Comparison with Other Spatial Question-Answering Benchmarks (Section D):** In this section, we compare MMSI-Bench with existing spatial question-answering benchmarks, highlighting the unique value of our benchmark.

4. **Additional Implementation Details and Experimental Results (Section E):** This section provides supplementary technical details regarding the implementation as well as more experimental results.

5. **Details of the Data Curation Process (Section F):** This section elaborates on the procedures and criteria used in curating the MMSI-Bench dataset, including data sourcing, annotation protocols, and quality control measures, as well as the methodology and results of the difficulty-level annotation.

6. **Representative MMSI-Bench Samples from Each Category (Full Version) (Section G):** In this section, we present a full set of representative samples from each category in MMSI-Bench, without any truncation or simplification. This is intended to give the reader a complete view of the dataset's diversity and coverage.

7. **Additional Case Studies (Section H):** We provide further case studies to illustrate the model's reasoning process on MMSI-Bench, supplementing those already discussed in the main text.

Each section is intended to provide additional transparency and insight, ensuring the reproducibility and broader contextualization of our research.

## B    LLM USAGE, LIMITATIONS, SOCIAL IMPACT, AND LICENSE & ACCESS

### B.1    LLM USAGE

In the preparation of this manuscript, Large Language Models (LLMs) were used as a writing-assistance tool. The primary purpose of their use was to aid in polishing the language, which included improving grammar, refining sentence structure, and enhancing the overall clarity and readability of the text. All core research concepts, methodologies, analyses, and conclusions were conceived and formulated exclusively by the authors. The authors have reviewed the final text and take full responsibility for the entire content of this paper, ensuring its scientific accuracy and originality.

### B.2    LIMITATIONS

To ensure higher data quality and richer content, we chose to manually annotate question–answer pairs. This approach may limit scalability because it requires substantial human effort, which we acknowledge as a potential limitation of MMSI-Bench. Nevertheless, MMSI-Bench's current scale (1,000 samples) is sufficient for meaningful assessment, given that today's MLLMs still have substantial headroom for improvement on this benchmark. Its size is also comparable to widely used VQA benchmarks, such as MMStar (1,500 samples), MM-Vet (218 samples), and LLaVA-in-the-Wild (60 samples). When performance begins to saturate, a larger benchmark may indeed be necessary, but reaching that point will likely require new research breakthroughs or more human annotation effort.

Notably, MMSI-Bench's scale is competitive among spatial benchmarks. For reference: ERQA has 400 samples, MuriBench 478, UniQA-3D 500, SpatialVLM 546, BLINK 572, LEGO-Puzzle 1,100,

SpatialRGPT 1,406, and VSI-Bench 5,000. MMSI-Bench contains 1,000 samples, of which only the template-based VSI-Bench is substantially larger.

## B.3 BROADER IMPACT

The introduction of MMSI-Bench brings both potential benefits and risks to the research community and broader society. On the positive side, MMSI-Bench provides a challenging and comprehensive benchmark specifically designed for evaluating the spatial reasoning capabilities of vision-language models. By facilitating more nuanced and rigorous assessment, MMSI-Bench may help drive advances in embodied AI, assistive technologies, spatial robotics, and other domains where spatial understanding is critical.

However, we also recognize several risks and challenges associated with the development and release of such datasets. First, improvements in spatial reasoning enabled by MMSI-Bench may have unintended applications, such as enhanced surveillance or automated decision-making systems that could impact privacy or disproportionately affect certain groups. We encourage users of MMSI-Bench to adhere strictly to ethical guidelines and to consider the societal implications of deploying models benchmarked using our dataset.

From an environmental perspective, we acknowledge the significant computational resources required to evaluate modern vision-language models. By releasing MMSI-Bench and its associated scripts publicly, we aim to promote reproducibility and reduce redundant efforts, thereby minimizing unnecessary environmental impact.

In summary, while MMSI-Bench is intended to advance responsible and transparent research in spatial reasoning for vision-language models, we urge the community to remain vigilant regarding both its positive and negative societal impacts.

## B.4 ETHICS STATEMENT

As this research uses only publicly accessible pre-trained models, it does not present any ethical concerns. All procedures strictly conform to the guidelines established by the ICLR Code of Ethics.

## B.5 REPRODUCIBILITY STATEMENT

Our experiments follow the baseline configurations established by prior evaluation benchmarks or the original testing protocols of each respective model. Detailed implementation information is provided in the Evaluation Setup and Appendix sections. We will release all data and code, along with thorough documentation, to facilitate faithful reproduction of our main experimental findings. All aspects of this research are fully compliant with the ICLR Reproducibility Requirements.

## B.6 LICENSE AND ACCESS

The MMSI-Bench dataset is distributed under the Creative Commons Attribution 4.0 International License (CC BY 4.0). For all images included in the dataset, their original licensing terms (dtu; sca; nus; hab; ego; agi; dav; way) remain fully in effect.

We release MMSI-Bench under the CC-BY license and Terms of Use, and require that any use of the dataset for model evaluation be properly disclosed. This license supplements but does not override the original licenses of source materials; users must also comply with all relevant legal requirements concerning data subjects. This statement clarifies the obligations and liabilities associated with using this benchmark. While we strive to ensure the accuracy and legality of all samples, we do not guarantee their absolute completeness or correctness. We assume no responsibility for any legal or other issues that may arise from the use of MMSI-Bench, including but not limited to copyright infringement, privacy violations, or the misuse of sensitive information.

By accessing, downloading, or using MMSI-Bench, you acknowledge that you accept this statement and agree to comply with the full terms of the CC-BY license. If you do not agree with these terms or the CC-BY license, you are not permitted to use this benchmark. MMSI-Bench will be hosted and maintained on GitHub and the Hugging Face Hub platforms.

## C    MOTIVATION FOR MULTI-IMAGE INPUT

We expect that general spatial intelligence will, like humans, operate on video inputs. However, we start with a multi-image setting for three reasons:

**Foundational to video.** Multi-image understanding is the basis of video understanding. Analogous to SLAM, a system reconstructs a scene and estimates camera pose over a stream, but the front-end module (VO) still estimates the delta pose between consecutive frames before constructing a holistic perception of the scene. Given current limitations of MLLMs in spatial understanding, models must first handle multiple images before they can reliably understand full videos.

**Clearer diagnosis.** Multiple images allow more systematic analysis of current MLLMs' failure modes. With videos, if a model answers incorrectly, it is hard to localize the cause, e.g., long-video limits, image retrieval over long sequences, or spatial reasoning itself. The multi-image setup is more spatially focused and easier for error analysis, which we believe accelerates community progress.

**Relevant proxy for video.** Multi-image benchmarks also reflect capabilities needed for video-based systems. Our hypotheses are: (i) although humans learn spatial intelligence from videos, they can solve MMSI-Bench, suggesting our tasks capture a necessary condition for general (video-based) spatial intelligence; and (ii) a multi-image setting can be viewed as a multi-frame, low-FPS special case of video.

We hope MMSI-Bench will inspire future video-based benchmarks.

## D    COMPARISON WITH OTHER SPATIAL QUESTION-ANSWERING BENCHMARKS

Table 5: Comparison of spatial question-answering benchmarks. Some benchmarks contain QA pairs outside spatial understanding; the number of spatial QA is shown in parentheses. Human-AI performance gap is defined as the accuracy difference between the best-performing model reported in the corresponding paper and human performance; for benchmarks without reported human results, the gap is calculated with respect to 100% accuracy and is marked as "<value%". Note: Scenario entries labeled as "Indoor & Outdoor" or "Indoor" refer to real-world images.

| Benchmark | Annotation Method | Visual Input Modality | Number of QAs | Scenario | Human-AI Gap |
|---|---|---|---|---|---|
| VSR (Liu et al., 2023) | Auto & Human | Single-Image | 10K | Indoor & Outdoor | 25.0% |
| BLINK (Fu et al., 2024) | Auto & Human | Single/Multi-Image | 3,807 (572) | Real-world & Synthetic | 44.4% |
| UniQA-3D (Zuo et al., 2024) | Auto & Human | Single/Multi-Image | 500 | Real-world & Synthetic | 14.2% |
| MuriBench (Wang et al., 2024a) | Auto & Human | Multi-Image | 2,600 (478) | Real-world & Synthetic | 25.2% |
| SpatialVLM (Chen et al., 2024a) | Auto & Human | Single-Image | 546 | Indoor & Outdoor | <30% |
| SpatialRGPT (Cheng et al., 2024) | Auto | Single-Image | 1,406 | Real-world & Synthetic | <42% |
| MMIU (Meng et al., 2024) | Auto | Multi-Image | 11K (2904) | Real-world & Synthetic | <45% |
| ERQA (Team et al., 2025b) | Human | Single/Multi-Image | 400 | Indoor & Outdoor | <35% |
| LEGO-Puzzle (Tang et al., 2025) | Human | Single/Multi-Image | 1K | Synthetic | 39.5% |
| VSI-Bench (Yang et al., 2024) | Auto & Human | Video | 5K | Indoor | 33% |
| MMSI-Bench | Human | Multi-Image | 1,000 | Indoor & Outdoor | 55.3% |

Despite significant advances in MLLMs, robust evaluation of their spatial intelligence—especially in multi-image, real-world scenarios—remains limited. As summarized in Table 5, most existing benchmarks focus narrowly on single-image spatial understanding or offer only limited multi-image coverage as minor components within more general VQA suites (e.g., BLINK (Fu et al., 2024), MuriBench (Wang et al., 2024a)), rather than serving as dedicated spatial reasoning benchmarks. Furthermore, several datasets rely heavily on template- or rule-based question generation (Ray et al., 2024; Zuo et al., 2024; Meng et al., 2024; Yang et al., 2024), which restricts both the diversity and depth of spatial reasoning challenges. Synthetic settings such as LEGO-Puzzle (Tang et al., 2025) further lack the complexity and contextual richness of real-world environments.

In contrast, MMSI-Bench is a fully human-curated benchmark, meticulously annotated by six 3D vision researchers without the use of templates. It offers comprehensive coverage of spatial relationships and elements—including camera (agent), objects, and regions—across ten well-defined task types plus one multi-step reasoning category. All scenarios derive from diverse, real-world sources, spanning indoor 3D scene datasets, local scene reconstructions, autonomous driving datasets, and robotics data, encompassing both indoor and outdoor environments. Notably, MMSI-Bench

Table 6: Quantitative comparison of question diversity against major non-single-image spatial reasoning benchmarks. Distinct words: Counts the total number of unique words after lowercasing and tokenization across all questions, measuring vocabulary richness. Unique n-grams: Counts the number of unique consecutive n-word sequences (here, n=3 and 5) after tokenization, measuring phrasal diversity. Syntactic structures: Calculated by parsing the syntax of each sentence, abstracting it into a tree structure by removing specific words, and then counting the number of unique structures. This measures the diversity of sentence constructions.

| Statistic | VSI-Bench | MMSI-Bench | ERQA (Multi-Img) | LEGO-Puzzle |
|---|---|---|---|---|
| Distinct words | 398 | **1,108** | 559 | 275 |
| Unique 3-grams | 4,482 | **8,499** | 2,483 | 1,023 |
| Unique 5-grams | 12,972 | **13,010** | 3,318 | 1,373 |
| Syntactic structures | 473 | **993** | 310 | 51 |

provides detailed, human-authored reasoning chains for each question, enabling rigorous assessment of both answer accuracy and reasoning quality. This benchmark reveals a substantial gap between current MLLMs and human-level spatial intelligence, and sets a higher standard for multi-image spatial reasoning evaluation in realistic settings.

To further quantify the linguistic diversity of MMSI-Bench, we conduct a comparative analysis against representative non-single-image spatial reasoning benchmarks, including VSI-Bench, the multi-image subset of ERQA, and LEGO-Puzzle. We analyze the complexity of the questions across these datasets at the vocabulary, phrase, and syntactic levels. The results are summarized in Table 6.

The analysis demonstrates that MMSI-Bench exhibits superior diversity across all metrics. At the vocabulary level, MMSI-Bench features 1,108 distinct words, significantly more than the other benchmarks. At the phrase level, it leads in the number of unique 3-grams and 5-grams, suggesting a wider variety of phrasal combinations and expressions. Most notably, at the syntactic level, MMSI-Bench contains 993 unique syntactic structures—nearly double that of VSI-Bench and more than triple that of ERQA. This linguistic and structural complexity stems from our fully human-curated annotation process, which is driven by experts and avoids any reliance on templates. This ensures that models are challenged not only with spatial reasoning tasks but also with understanding and processing diverse, naturally-phrased questions that reflect real-world linguistic distribution. Consequently, it sets a higher bar for evaluating a model's generalization capabilities.

# E ADDITIONAL IMPLEMENTATION DETAILS AND EXPERIMENTAL RESULTS

## E.1 BENCHMARK MODELS.

We conduct a comprehensive evaluation of a wide range of MLLMs, encompassing both proprietary and open-source families, diverse model scales, and recent architectural advances. For proprietary models, we includes GPT-5 (OpenAI, 2025c), o3 (OpenAI, 2025d), GPT-4.5 (OpenAI, 2025b), GPT-4.1 (OpenAI, 2025a), GPT-4o (OpenAI, 2024), Gemini-2.5-Pro (DeepMind, 2025), Claude-3.7-Sonnet (Anthropic, 2025), dots.vlm 1 (Huo et al., 2025), GLM-4.5V-thinking (Hong et al., 2025), Doubao-1.5-pro (Seed, 2025), and Seed1.5-VL (Guo et al., 2025). On the open-source side, we systematically assess state-of-the-art models such as Qwen2.5-VL (Bai et al., 2025), LLaVA-OneVision (Li et al., 2024), InternVL3 (Zhu et al., 2025), InternVL2.5 (Chen et al., 2024b), DeepSeek-VL2 (Wu et al., 2024), Llama-3.2-11B-Vision (Meta, 2024), and NVILA (Liu et al., 2024c).

## E.2 IMPLEMENTATION DETAILS FOR MODEL EVALUATION

We report accuracy (%) using exact match between answers extracted from model outputs and ground-truth answers for our multiple-choice questions. If a model fails to generate an answer in the required format, we adopt the LLM-based fallback strategy from VLMEvalKit (Duan et al., 2024) to extract the intended response.

We evaluate models using three prompting strategies: Direct, Linguistic (Zero-Shot Chain-of-Thought), and Visual. The exact prompts used for each method are shown in Table 7.

Table 7: Prompts used for different evaluation settings.

| Strategy | Prompt Template |
|---|---|
| Direct | {images}{question}
Answer with the option's letter from the given choices directly. Enclose the option's letter within ` `. |
| Linguistic (Zero-Shot CoT) | {images}{question}
Let's think step by step and then answer with the option's letter from the given choices. Enclose the option's letter within ` `. |
| Visual | {image with visual prompt}{question}
Answer with the option's letter from the given choices directly. Enclose the option's letter within ` `. |

For the visual prompting strategy, "{image with visual prompt}" refers to the image input combined with a visual prompt drawn using PATS (Ni et al., 2023), as illustrated in Figs. 8, 9, 10, 11.

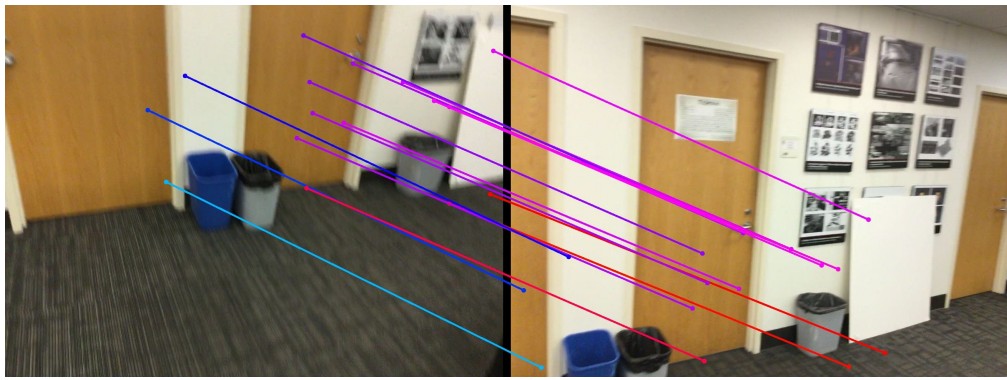

Figure 8: Illustration of the visual prompt utilized in our experiments.

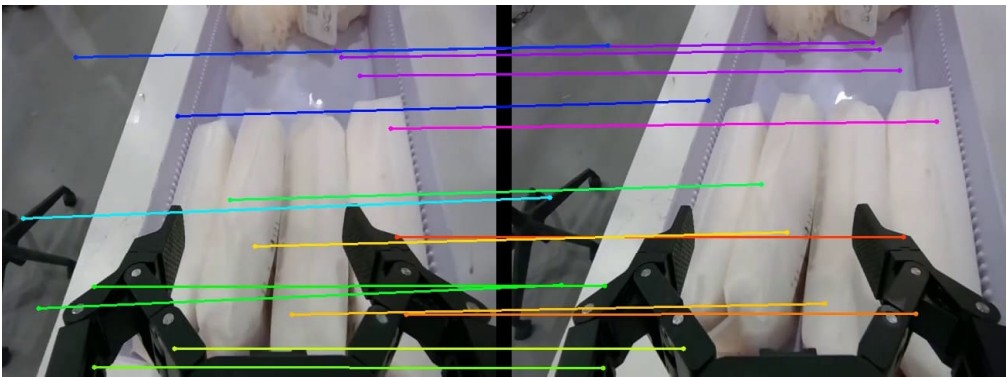

Figure 9: Illustration of the visual prompt utilized in our experiments.

The specific prompt used for evaluating the reasoning process is shown in Table 8.

The prompt for evaluating reasoning processes with a human-annotated reference is shown in Table 9.

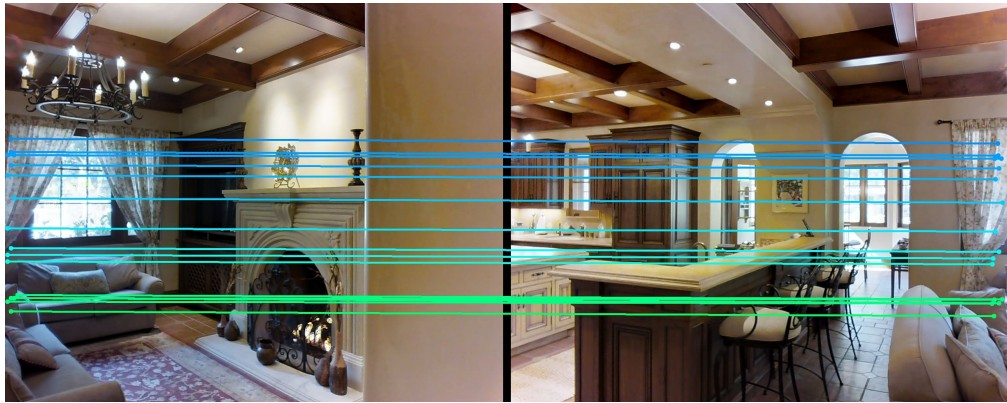

Figure 10: Illustration of the visual prompt utilized in our experiments.

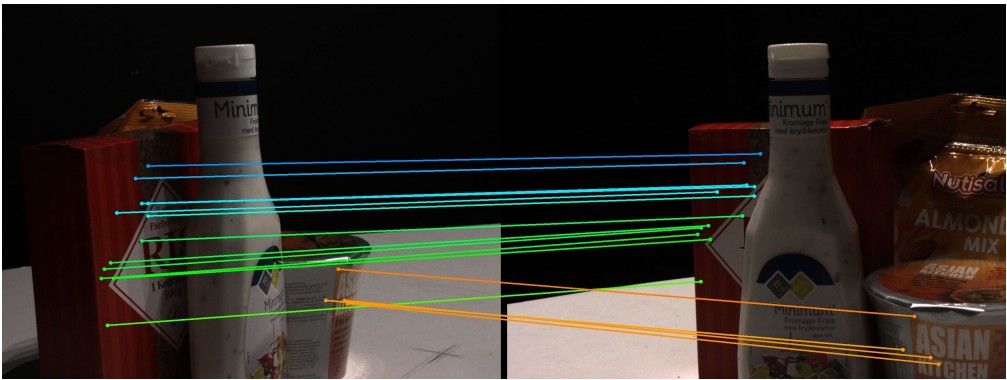

Figure 11: Illustration of the visual prompt utilized in our experiments.

Table 8: Prompt used for evaluating the reasoning process in our study.

| Prompt |
| --- |
| {images}{question} |

{images}{question}
{reasoning process to be evaluated}
{the correct answer}
You are given images, a question about the images, the correct answer, and a reasoning process to be evaluated.
Your task is to evaluate the reasoning process and determine whether it is entirely correct. If there are any errors, identify the type of error that most critically led to the incorrect answer by selecting the corresponding category number. If the reasoning process is completely correct, select only category 5.
Categories:
1. Grounding error: Failure to accurately detect, locate, or reference objects or elements in the image.
2. Overlap identification and scene reconstruction error: Mistakes in identifying overlaps between elements or errors in reconstructing spatial relationships within the scene.
3. Situation transformation reasoning error: Errors in reasoning about directions or spatial relations from the perspective of an entity other than the observer (i.e., allocentric transformation errors).
4. Spatial logic error: Flaws in spatial reasoning logic, such as fabrication, linguistic intelligence error, misusing reference objects in motion reasoning, or other errors not covered by categories 1, 2, or 3.
5. Completely correct reasoning.
Please output only the relevant category number.

Table 9: Prompt used for evaluating reasoning processes with human-annotated reference in our study. Here, the structured format of the reasoning process means that, based on the original human-annotated reasoning steps, each step is further labeled with its reasoning type. There are four reasoning types, Grounding, Overlap-Matching and Scene-Reconstruction, Situation-Transformation Reasoning, and Spatial-Logic Reasoning, which correspond to four types of possible errors, respectively. For example, a reasoning process can be transformed into the structured format as follows: "The pink bench is to the east of the bed. The white wall lies to the south. Thus, the bathroom lies in the southeast corner of the sleeping region." will become "The pink bench is to the east of the bed (Grounding). The white wall lies to the south (Grounding). Thus, the bathroom lies in the southeast corner of the sleeping region (Overlap-Matching and Scene-Reconstruction)."

---

**Prompt**

---

{images}{question}
{reasoning process to be evaluated}
{the correct answer}
{human-annotated reference reasoning process with structured format}
You are given images, a question about the images, the correct answer, a human-annotated reference reasoning process, and a reasoning process to be evaluated.
Your task is to evaluate the reasoning process and determine whether it is entirely correct. If there are any errors, identify the type of error that most critically led to the incorrect answer by selecting the corresponding category number. If the reasoning process is completely correct, select only category 5.
Categories:
1. Grounding error: Failure to accurately detect, locate, or reference objects or elements in the image.
2. Overlap identification and scene reconstruction error: Mistakes in identifying overlaps between elements or errors in reconstructing spatial relationships within the scene.
3. Situation transformation reasoning error: Errors in reasoning about directions or spatial relations from the perspective of an entity other than the observer (i.e., allocentric transformation errors).
4. Spatial logic error: Flaws in spatial reasoning logic, such as fabrication, linguistic intelligence error, misusing reference objects in motion reasoning, or other errors not covered by categories 1, 2, or 3.
5. Completely correct reasoning.
Note: The human-annotated reference reasoning process is not the only correct answer and is for reference only. Other reasoning paths may also be correct. You should judge the correctness of the reasoning process and the error category based on the specific images.
Please output only the relevant category number.

---

### E.3 HUMAN EVALUATION SETUP

Human performance is measured as the average accuracy of five adult participants who were not involved in the data annotation process. During evaluation, human annotators are provided with both the questions and the corresponding images simultaneously and are allowed unlimited time to answer each question to the best of their ability. They may review the images as many times as needed to gather comprehensive information. The evaluation interface supports two modes of image viewing: all images can be displayed side by side for direct comparison, or the participant can switch between individual images as needed. This flexible setup ensures that evaluators can fully utilize the available visual information for accurate reasoning and decision-making.

### E.4 COMPLETE EVALUATION RESULTS OF PROMPTING TECHNIQUES ON MMSI-BENCH

We provide the complete evaluation results of prompting techniques on MMSI-Bench in Table 10, which correspond to the results presented in Figure 5 of the main text.

### E.5 CORRELATION WITH DOWNSTREAM EMBODIED AND ACTIVE PERCEPTION TASKS

To assess the transferability of MMSI-Bench to downstream embodied and active perception tasks, we analyze the correlation between model performance on MMSI-Bench and three representative tasks from EMBODIEDBENCH Yang et al. (2025): EB-Navigation, EB-Manipulation, and EB-Habitat.

Table 10: Complete evaluation results of prompting techniques on MMSI-Bench.

| Models | Positional Relationship | | | | | | Attribute | | Motion | | MSR | Avg. |
|---|---|---|---|---|---|---|---|---|---|---|---|---|
| | Cam.–Cam | Obj.–Obj. | Reg.–Reg. | Cam.–Obj. | Obj.–Reg. | Cam.–Reg. | Meas. | Appr. | Cam. | Obj. | – | |
| *The original MMSI-Bench setting without the use of any prompting techniques.* | | | | | | | | | | | | |
| GPT-4.5 | 34.4 | 29.8 | 39.5 | 51.2 | 47.1 | 55.4 | 39.1 | 33.3 | 41.9 | 40.8 | 36.4 | 40.3 |
| GPT-4o | 34.4 | 24.5 | 23.5 | 19.8 | 37.6 | 27.7 | 32.8 | 31.8 | 35.1 | 36.8 | 30.8 | 30.3 |
| Qwen2.5-VL-72B | 25.8 | 34.0 | 34.6 | 23.3 | 34.1 | 36.1 | 45.3 | 27.3 | 27.0 | 30.3 | 27.3 | 30.7 |
| LLaVA-OneVision-72B | 43.0 | 31.9 | 33.3 | 30.2 | 37.6 | 38.6 | 28.1 | 19.7 | 13.5 | 32.9 | 15.7 | 28.4 |
| *Linguistic prompting* | | | | | | | | | | | | |
| GPT-4.5 | 44.1 | 28.7 | 35.8 | 32.6 | 40.0 | 43.4 | 57.8 | 31.8 | 41.9 | 36.8 | 42.4 | 39.6 |
| GPT-4o | 42.1 | 24.6 | 22.7 | 25.9 | 34.5 | 37.8 | 48.0 | 35.9 | 30.4 | 29.6 | 28.8 | 32.0 |
| Qwen2.5-VL-72B | 28.9 | 30.8 | 34.5 | 32.5 | 24.6 | 30.0 | 40.5 | 28.7 | 22.9 | 31.5 | 23.6 | 29.0 |
| LLaVA-OneVision-72B | 31.7 | 24.0 | 26.5 | 31.3 | 22.1 | 27.3 | 25.2 | 20.1 | 13.4 | 8.0 | 23.2 | 24.7 |
| *Concatenating multiple images into a single composite image without using any prompting techniques.* | | | | | | | | | | | | |
| GPT-4.5 | 35.5 | 30.8 | 34.6 | 33.7 | 43.5 | 28.9 | 43.8 | 27.3 | 32.4 | 26.3 | 28.3 | 32.6 |
| GPT-4o | 28.0 | 22.3 | 38.3 | 24.4 | 41.2 | 31.3 | 32.8 | 19.7 | 21.6 | 30.3 | 26.3 | 28.5 |
| Qwen2.5-VL-72B | 36.6 | 22.3 | 34.6 | 18.6 | 35.3 | 28.9 | 37.5 | 16.7 | 23.0 | 31.6 | 31.3 | 29.1 |
| LLaVA-OneVision-72B | 32.3 | 24.5 | 30.9 | 26.7 | 37.6 | 27.7 | 32.8 | 19.7 | 9.5 | 26.3 | 29.8 | 27.6 |
| *Visual prompting* | | | | | | | | | | | | |
| GPT-4.5 | 36.6 | 31.9 | 30.9 | 31.4 | 42.4 | 27.7 | 37.5 | 31.8 | 24.3 | 28.9 | 30.8 | 32.1 |
| GPT-4o | 28.7 | 29.5 | 28.8 | 25.0 | 37.0 | 30.6 | 40.4 | 27.1 | 16.4 | 30.5 | 29.2 | 29.3 |
| Qwen2.5-VL-72B | 30.9 | 27.4 | 30.4 | 28.7 | 36.1 | 32.1 | 30.5 | 25.0 | 19.7 | 29.7 | 27.1 | 28.8 |
| LLaVA-OneVision-Qwen2-72B | 29.0 | 33.0 | 29.6 | 31.4 | 38.8 | 33.7 | 29.7 | 21.2 | 14.9 | 26.3 | 28.3 | 29.0 |

- **EB-Navigation** evaluates embodied agents' navigation abilities in simulated environments, where models are required to output low-level actions to reach target locations.

- **EB-Manipulation** focuses on object-centric embodied manipulation skills, such as grasping or rearranging objects, and requires fine-grained spatial reasoning and interaction.

- **EB-Habitat** is built on the Habitat 2.0 simulator and targets high-level active perception and planning: agents must plan and execute sequences of skills to visit multiple locations and find user-specified items.

**Correlation analysis.** We compute Pearson and Spearman correlation coefficients between the average accuracy on MMSI-Bench and performance on each downstream task across a diverse set of frontier multimodal models. The results show consistently strong positive correlations:

- **MMSI-Bench → EB-Navigation:** Pearson = 0.8147,   Spearman = 0.7333.

- **MMSI-Bench → EB-Manipulation:** Pearson = 0.7402,   Spearman = 0.7500.

- **MMSI-Bench → EB-Habitat:** Pearson = 0.7299,   Spearman = 0.6167.

These results indicate that models performing well on MMSI-Bench also tend to achieve strong performance on both downstream embodied control (navigation, manipulation) and active perception (navigation, Habitat) tasks. The particularly high correlation with EB-Navigation suggests that the spatial understanding and scene interpretation abilities measured by MMSI-Bench are closely aligned with the competencies required for effective embodied navigation. At the same time, the substantial correlations with EB-Manipulation and EB-Habitat demonstrate that MMSI-Bench captures capabilities that generalize to both low-level control and high-level, goal-directed active perception.

**Per-model performance.** Table 11 presents the detailed per-model scores on MMSI-Bench and the three EMBODIEDBENCH tasks that underlie the above correlation analysis.

# F   DETAILS OF THE DATA CURATION PROCESS

## F.1   DATASET SOURCES

**ScanNet (Dai et al., 2017):** ScanNet is a large-scale RGB-D video dataset designed for 3D scene understanding in real-world indoor environments. It consists of 2.5 million RGB-D images across 1,513 scans from 707 unique spaces, each annotated with 3D camera poses, surface reconstructions,

Table 11: Performance of different models on MMSI-Bench and three downstream tasks from EMBODIEDBENCH: EB-Navigation, EB-Manipulation, and EB-Habitat. All values are reported as percentages.

| Model | MMSI-Bench | EB-Navigation | EB-Manipulation | EB-Habitat |
|---|---|---|---|---|
| GPT-4o | 30.3 | 57.7 | 39.6 | 59.0 |
| Claude-3.7-Sonnet | 30.2 | 45.0 | 28.5 | 58.7 |
| Claude-3.5-Sonnet | 31.3 | 44.7 | 25.4 | 68.0 |
| Llama-3.2-11B-Vision-Ins | 25.4 | 21.4 | 0.9 | 25.0 |
| InternVL2.5-78B | 28.5 | 30.7 | 10.6 | 49.0 |
| InternVL2.5-38B | 27.9 | 30.3 | 15.8 | 38.3 |
| InternVL2.5-8B | 28.7 | 21.3 | 7.0 | 11.3 |
| Qwen2.5-VL-72B-Ins | 30.7 | 40.0 | 16.2 | 37.7 |
| Qwen2.5-VL-7B-Ins | 25.9 | 20.0 | 9.6 | 14.3 |

and dense object-level semantic segmentations. The dataset was collected using a scalable pipeline that combines easy data capture with crowdsourced annotations, making it one of the most richly-labeled and comprehensive resources for training and evaluating deep learning models on tasks such as 3D object classification and semantic voxel labeling.

**nuScenes (Caesar et al., 2020):** nuScenes is a multimodal autonomous driving dataset that features a complete sensor suite, including 6 cameras, 5 radars, and 1 lidar, all with 360-degree coverage. It contains 1,000 driving scenes, each 20 seconds long, fully annotated with 3D bounding boxes for 23 object classes and 8 attributes. nuScenes stands out for its diversity of sensor data, large scale of annotations, and support for advanced tasks such as 3D detection and tracking, making it a benchmark for research in perception for autonomous vehicles.

**Matterport3D (Chang et al., 2017):** Matterport3D is a large-scale RGB-D dataset capturing 10,800 panoramic viewpoints from 194,400 images of 90 building-scale indoor scenes. It provides surface reconstructions, globally aligned camera poses, and detailed 2D and 3D semantic segmentations at both region and object levels. The comprehensive panoramic coverage and precise global alignment enable a wide range of computer vision tasks, including keypoint matching, semantic segmentation, and view overlap prediction, particularly in the context of indoor scene understanding.

**Ego4D (Grauman et al., 2022):** Ego4D is an unprecedented egocentric video dataset, comprising 3,670 hours of first-person footage collected by 931 participants across 74 locations in 9 countries. The dataset covers a wide array of daily-life scenarios and includes supplementary data such as audio, 3D meshes, gaze, and multi-camera views. With millions of annotations supporting various benchmark tasks, Ego4D enables research into first-person perception, episodic memory, and social interactions, significantly advancing the field of egocentric visual understanding.

**AgiBot-World (AgiBot-World-Contributors et al., 2025):** AgiBot-World is a large-scale robotic manipulation dataset featuring over 1 million trajectories across 217 tasks in five real-world domains—including domestic, retail, industrial, restaurant, and office environments—collected by more than 100 real robots. Data collection is standardized and includes human-in-the-loop verification to ensure high quality and diversity. The dataset supports a wide range of hardware, from simple grippers to dexterous hands with visuo-tactile sensors, and provides rich multimodal annotations. AgiBot-World enables the development and evaluation of generalist robot policies in complex, long-horizon, and diverse scenarios, moving towards scalable, general-purpose robotic intelligence.

**DTU (Jensen et al., 2014):** The DTU dataset is a large-scale multi-view stereo (MVS) benchmark designed for 3D reconstruction research. It contains 80 scenes with significant variability in surface properties and geometric complexity. Each scene is captured from 49 or 64 accurate camera positions and includes high-quality reference scans obtained via structured light. This dataset advances the scale and diversity compared to previous MVS datasets and serves as a challenging benchmark for evaluating the accuracy and completeness of stereo reconstruction algorithms.

**DAVIS 2017 (Pont-Tuset et al., 2017):** DAVIS 2017 is a benchmark dataset for video object segmentation, consisting of 150 video sequences and a total of 10,459 annotated frames with pixel-level masks for multiple objects per scene. Compared to earlier versions, DAVIS 2017 introduces

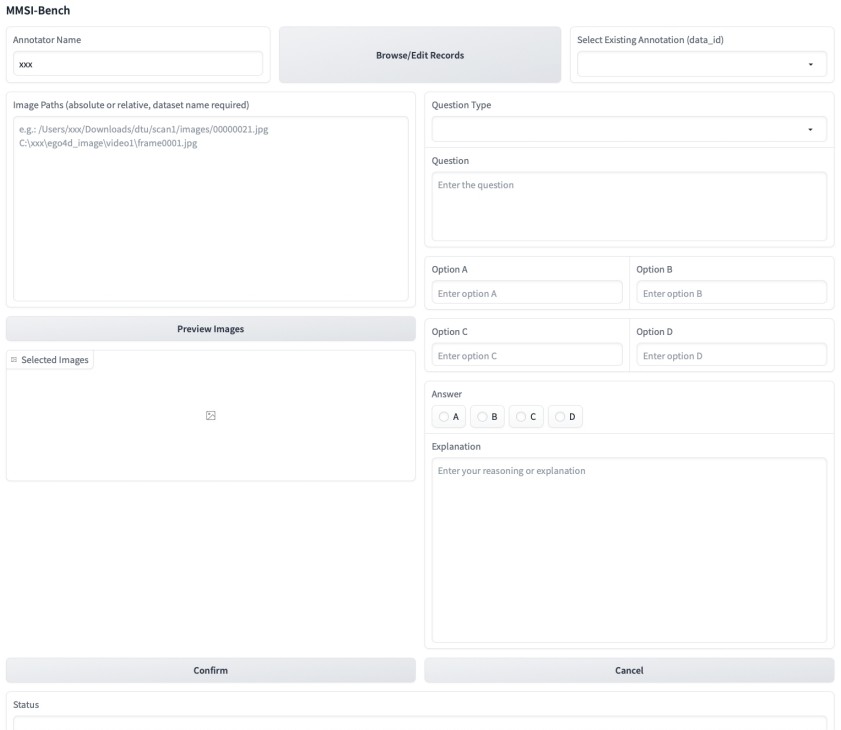

Figure 12: Screenshot of the data annotation interface, where annotators select images, design questions, and provide answers and reasoning.

greater complexity, including multiple objects, occlusions, small and fine structures, and fast motion. The dataset has driven significant progress in video object segmentation and is widely used for benchmarking and competition in the field.

**Waymo Open Dataset (Sun et al., 2020):** The Waymo Open Dataset is one of the largest and most diverse multimodal datasets for autonomous driving. It comprises 1,150 scenes, each lasting 20 seconds, captured across multiple cities with extensive geographic coverage. Each scene includes synchronized and calibrated high-resolution camera images and LiDAR data, along with exhaustive 2D and 3D bounding box annotations and consistent object tracking IDs. The dataset supports research in detection, tracking, and sensor fusion, providing a valuable foundation for developing robust and generalizable self-driving technologies.

MMSI-Bench is composed of questions from 8 data sources: 463 from Matterport3D, 280 from ScanNet, 67 from Ego4D, 45 from AgiBot-World, 72 from DTU, 39 from nuScenes, 29 from DAVIS 2017, and 5 from Waymo. We analyzed widely used VLM pre-training datasets, CC3M (Changpinyo et al., 2021) and LAION-COCO (Schuhmann et al., 2022), which are commonly utilized by open-source VLMs such as LLaVA, NVILA, and Qwen-VL. Our analysis indicates that outdoor scenes constitute only an estimated 10% of the data in these foundational VLM datasets. To align our benchmark with the typical training distribution of these models, we have curated MMSI-Bench to maintain a comparable proportion, with approximately 12% of our samples sourced from outdoor datasets. Recognizing that spatial understanding is one of the most critical capabilities for autonomous driving, we selected Waymo and nuScenes as the primary sources for these outdoor samples.

## F.2 DATA ANNOTATION AND QUALITY CONTROL INTERFACES

The interfaces for data annotation and quality control are illustrated in Figure 12 and Figure 13, respectively. During annotation, annotators select an image, design a question, provide the correct answer, and write the corresponding reasoning process. For quality control, reviewers can choose to discard a data entry if any issues are identified, providing the reason for rejection; otherwise, they can approve and retain the data.

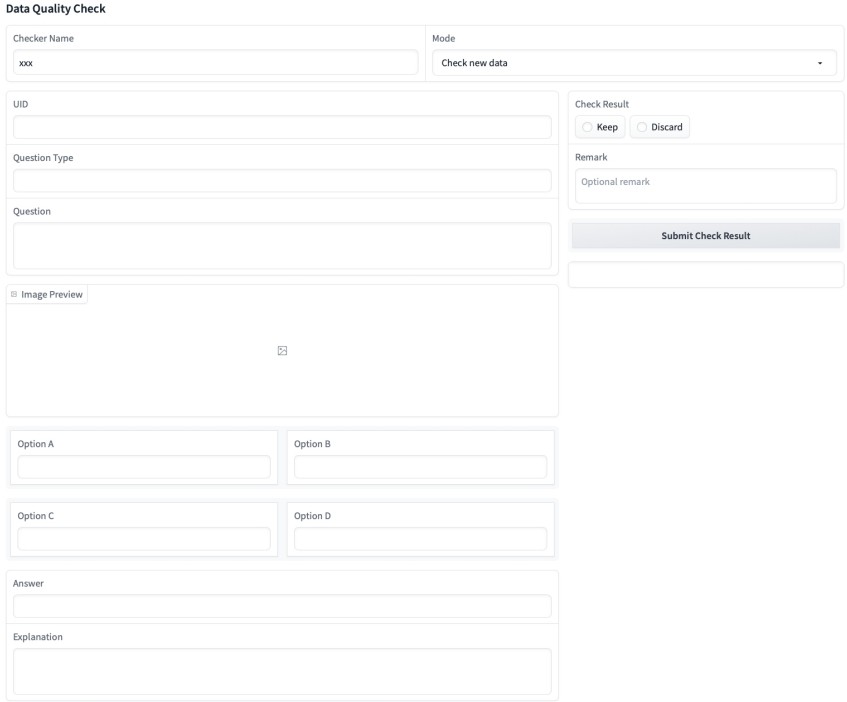

Figure 13: Screenshot of the quality control interface, where reviewers decide whether to retain or discard data and specify the reasons for data issues.

### F.3 GENERAL GUIDELINES

MMSI-Bench is designed to systematically evaluate the multi-image spatial reasoning capabilities of multimodal large language models (MLLMs) using human-readable questions. The following guidelines govern the data annotation process:

- **Human-Centric Design:** All questions and answer choices are formulated in natural, human-readable English, avoiding technical prompts such as direct requests for camera parameters.

- **Image Requirement:** Every question is constructed so that it *cannot* be answered using a single image alone; the information required to answer must be synthesized from multiple images.

- **Question Type and Format:** All questions are multiple-choice with four options, only one of which is correct. Questions are created in free form (no templates) to ensure diversity and challenge for MLLMs.

- **Category Coverage:** Each question is assigned to one of eleven categories, covering positional relationships, attributes, motion, and multi-step reasoning among cameras, objects, and regions. This taxonomy is designed to comprehensively span the space of fundamental spatial reasoning types.

- **Difficulty, Diversity, and Clarity:** Questions must be of high difficulty, unambiguous, and answerable by humans. Annotators are encouraged to maximize diversity in question design and avoid repetitive patterns. Annotations must be accurate, consistent, and adhere to a high standard of academic rigor.

- **Ethics and Licensing:** All images and question sources must comply with copyright and licensing restrictions. Annotators must not use material from sources that prohibit redistribution.

- **Reasoning Process Annotation:** For each question, annotators provide a detailed step-by-step reasoning process, documenting the logical path from the images to the answer.

For the "Multi-Step Reasoning" type, we ask annotators to design questions that are as challenging as possible and cannot be answered directly, and to explicitly write down the step-by-step solution in the reasoning process. According to our statistics on human-annotated reasoning processes in this category, "Multi-Step Reasoning" questions involve between 3 and 11 reasoning steps, with an average of 5.7 steps per question.

### F.4 DATA FORMAT AND STRUCTURE

To ensure consistency and facilitate reliable evaluation and analysis, all benchmark entries conform to a standardized data structure:

- **JSON Format:** Each question-answer pair is stored as a JSON object, with fields for question text, four answer options, the correct answer index, category, difficulty, image paths, and annotated reasoning process.
- **Image Files:** All referenced images are stored separately and linked via file paths in the JSON entries.
- **Field Completeness:** Each entry must have all fields completed: question, answer, category, image list, and reasoning process.
- **Multi-Image Structure:** Except for the Multi-Step Reasoning category, every question is associated with exactly two images. Multi-Step Reasoning samples may include more images according to task complexity.
- **Final Storage Format:** For efficient downloading and usage, the entire dataset—including binary-encoded images and metadata—is stored in the Parquet format.

### F.5 QUALITY CONTROL AND VALIDATION

A rigorous multi-stage quality control procedure is implemented to ensure the reliability and validity of the benchmark:

- **Independent Review:** All annotated samples are independently reviewed by three expert reviewers who were not involved in the original annotation process.
- **Error and Ambiguity Filtering:** Reviewers systematically remove or revise questions exhibiting insufficient visual information, ambiguity (linguistic or visual), incorrect answers, or vague/incomplete reasoning processes.
- **Non-Triviality Check:** Samples that can be answered using a single image or general common sense (rather than spatial reasoning across images) are excluded.
- **Iterative Feedback:** Annotators receive feedback on problematic items and are required to revise or replace such entries.
- **Final Audit:** Only questions passing all review stages are included in the benchmark, ensuring high data quality and robust evaluation.

### F.6 DIFFICULTY ANNOTATION

We recruited new human evaluators to answer MMSI-Bench and recorded their per-question completion times. Each question was independently assigned to two evaluators. We normalized completion times across evaluators and used the mean normalized time as the difficulty indicator, with longer times indicating higher difficulty. The resulting distribution exhibited a clear three-tier structure. We therefore labeled each question as Easy, Medium, or Hard. Any question answered incorrectly by the evaluators was directly assigned to the Hard tier. The Easy tier contains 605 questions with an average human completion time of approximately 40 seconds. The Medium tier contains 262 questions with an average time of approximately 110 seconds. The Hard tier contains 133 questions, defined as those taking longer than approximately 110 seconds on average or those answered incorrectly.

### F.7 DETAILS ABOUT THE DISTRIBUTION OF DATA ACROSS THE 11 TASKS

For the Motion (Cam.) task, we use a total of 74 samples: 32 from Ego4D, 27 from AgiBot-World, 8 from ScanNet, and 7 from nuScenes (combining all nuScenes image subsets).

For the Positional Relationship (Cam.–Obj.) task, we use 86 samples in total. Among them, 61 samples are drawn from Matterport3D (combining validation and test splits), and 25 samples are from ScanNet.

For the Attribute (Meas.) task, we use 84 samples: 40 from Matterport3D, 19 from ScanNet, and 5 from DTU.

For the Positional Relationship (Reg.–Reg.) task, we use 81 samples in total, of which 79 come from Matterport3D and 2 from ScanNet.

For the MSR task, we use 197 samples overall. Specifically, 96 samples are from ScanNet, 79 from Matterport3D, 10 from DAVIS 2017 (via the tapvid-based motion setup), 5 from AgiBot-World, 6 from nuScenes, and 2 from Waymo.

For the Motion (Obj.) task, we use 76 samples. These include 35 from Ego4D, 19 from DAVIS 2017, 13 from AgiBot-World, 6 from nuScenes, and 3 from Waymo.

For the Positional Relationship (Cam.–Cam.) task, we use 92 samples in total. This set consists of 60 samples from ScanNet, 20 from nuScenes, 11 from Matterport3D, and 2 from DTU.

For the Positional Relationship (Cam.–Reg.) task, we use 83 samples in total, including 37 from ScanNet and 46 from Matterport3D.

For the Attribute (Appr.) task, we use 66 samples. We draw 65 samples from DTU (aggregating all DTU variants we use) and 1 sample from ScanNet.

For the Positional Relationship (Obj.–Reg.) task, we use 85 samples in total, with 75 samples from Matterport3D and 10 from ScanNet.

Finally, for the Positional Relationship (Obj.–Obj.) task, we use 94 samples, composed of 72 samples from Matterport3D and 22 from ScanNet.

## G    REPRESENTATIVE MMSI-BENCH SAMPLES FROM EACH CATEGORY (FULL VERSION)

For clarity and readability, questions and rationales in the main text are simplified; the complete versions for each category are provided in this section. We present one full example for each of the 11 categories: Position (Camera-Camera), Position (Camera-Object), Position (Camera-Region), Position (Object-Object), Position (Object-Region), Position (Region-Region), Motion (Camera), Motion (Object), Attribute (Measurement), Attribute (Appearance), and Multi-Step Reasoning. These representative samples are illustrated in Figures 14–24.

## H    ADDITIONAL CASE STUDIES

In this section, we present additional complete reasoning processes of current multimodal large language models (MLLMs) to illustrate their spatial reasoning deficiencies more intuitively. As shown in Figure 25, Figure 26, Figure 27, and Figure 28, we highlight four types of errors in different colors: grounding errors (blue), overlap-matching and scene-reconstruction errors (red), situation-transformation reasoning errors (orange), and spatial-logic errors (green).

Position (Cam.-Cam.)

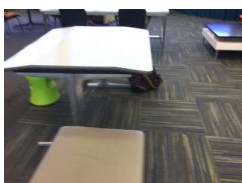 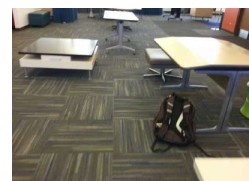

Question: Assuming I am taking the first image, where is the camera likely positioned relative to me when taking the second image?
Options: A: Left, B: Front, C: Back, D: Right

Reasoning: The first image captures the green chair and the backpack on the left and right sides, respectively. The second image captures the green chair and the backpack on the right and left sides, respectively, and it can be seen that the brown leather chair, which was very close to the camera in the first image, is farther from the camera and almost directly in front in the second image. It can be inferred that the approximate direction is forward.

Figure 14: Full example for Position (Camera-Camera) category.

Position (Cam.-Obj.)

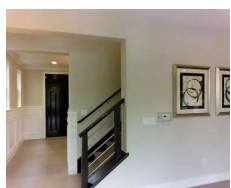 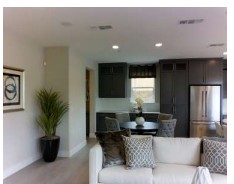

Question: When I took the second image, where was the black door in relation to me?
Options: A: Behind me, B: On my left, C: In front of me, D: On my right

Reasoning: The images have overlapping areas — the painting on the right side of Image 1 and the painting on the left side of Image 2 are the same. By aligning the overlapping parts, we can perform spatial reasoning. In Image 1, the painting is in front and to the right of the photographer, while the black door is directly in front. In Image 2, the painting is in front and to the left of the photographer. Therefore, from Image 1 to Image 2, the photographer rotated 90 degrees to the right. As a result, when taking the second image, the black door was on the photographer's left.

Figure 15: Full example for Position (Camera-Object) category.

Position (Cam.-Reg.)

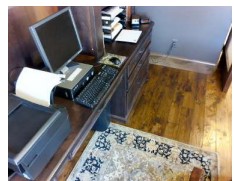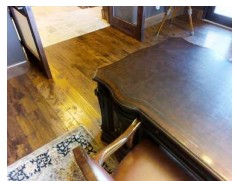

Question: The images are taken sequentially from a first-person perspective. At the moment of taking the last image, in which direction is the file printing area relative to you?
Options: A: Behind, B: In front, C: Right, D: Left

Reasoning: The images were taken sequentially, and there is an overlap in the carpet between Image 1 and Image 2. Based on the relative position of the carpet in both images, it can be inferred that the camera rotated approximately 60 to 90 degrees to the right. In Image 1, the file printing area is located in front and slightly to the left of the photographer, so after the camera rotates to the angle in Image 2, the file printing area ends up on the your left.

Figure 16: Full example for Position (Camera-Region) category.

Position (Obj.-Obj.)

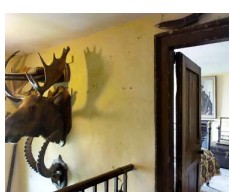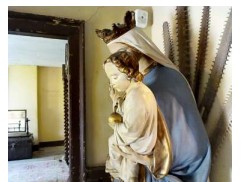

Question: The deer head ornament is on the west side of the mythological statue. In which direction from the mythological statue is the mural in the room?
Options: A: Southwest, B: Northeast, C: Northwest, D: Southeast

Reasoning: The door of the room on the right side of Image 1 and the door of the room on the left side of Image 2 have overlapping parts, so they belong to the same room. The deer head ornament is on the west of the mythological statue, and the mural in the room is further north than the deer head ornament. Therefore, the mural in the room is to the northwest of the mythological statue .

Figure 17: Full example for Position (Object-Object) category.

Position (Obj.-Reg.)

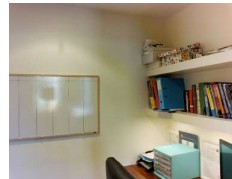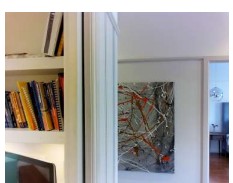

Question: The whiteboard in the study is hung on the north side of the study. In which direction is the painting on the wall relative to the center of the study?
Options: A: Northeast, B: Southeast, C: Northwest, D: Southwest

Reasoning: The books on the left side of Image 1 and the books on the right side of Image 2 have overlapping features. The whiteboard is on the north side of the study, so the bookshelf is on the east side. The painting is on the east side of the study, just like the bookshelf, but it is positioned further toward the south than the bookshelf. Therefore, the painting on the wall is in the southeast direction of the study.

Figure 18: Full example for Position (Object-Region) category.

Position (Reg.-Reg.)

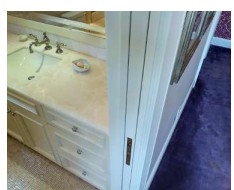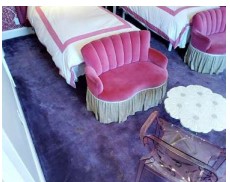

Question: The pink bench at the end of the bed is on the east side of the bed. In which direction is the bathroom located in the bedroom?
Options: A: Southwest, B: Northwest, C: Southeast, D: Northwest

Reasoning: The flooring and walls on the right side of Image 1 and the left side of Image 2 have overlapping features. The pink bench at the end of the bed is on the east side of the bed, so the white wall is on the south side of the bedroom. In Image 1, the bathroom is on the same side as the wall, which is the south side of the bedroom, but it is positioned further toward the east. Therefore, the bathroom is in the southeast direction of the bedroom.

Figure 19: Full example for Position (Region-Region) category.

Motion (Camera)

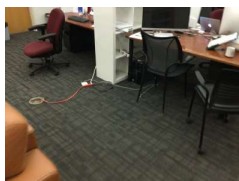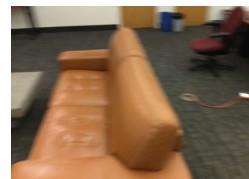

Question: The images are taken sequentially from a first-person perspective. In which direction is the camera rotating?
Options: A: Left, B: Right, C: Up, D: Down

Reasoning: The images were taken sequentially from a first-person perspective, and the orange sofa in the lower left corner of Image 1 also appears in Image 2. In Image 1, the sofa is on the left side of the photographer; in Image 2, the sofa is in front of the photographer. Therefore, from Image 1 to Image 2, the camera rotated to the left.

Figure 20: Full example for Motion (Camera) category.

Position (Cam.-Cam.)

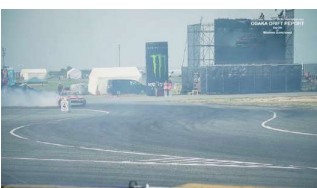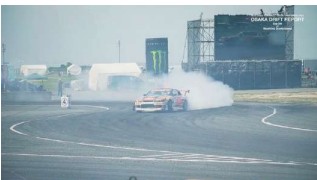

Question: With the camera facing backward and the images taken in sequence, in which direction is the race car moving?
Options: A: Left, B: Front left, C: Front right, D: Forward

Reasoning: The camera is facing backward, and the images were taken sequentially. In Image 2, the race car is closer to the photographer, indicating that it is moving forward. Compared to Image 1, it has shifted to the left, so from Image 1 to Image 2, the race car is moving toward the front-left.

Figure 21: Full example for Motion (Object) category.

Attribute (Measurement)

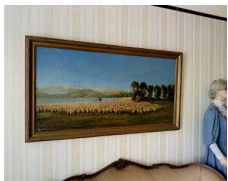 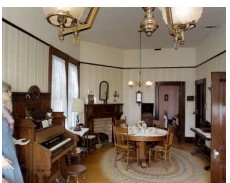

Question: Which is longer, the oil painting on the wall or the table below the mirror?
Options: A: Sometimes the table below the mirror is longer, sometimes the oil painting on the wall is longer, B: The oil painting on the wall, C: The same length, D: The table below the mirror

Reasoning: The right side of Image 1 and the left side of Image 2 have overlapping features, indicating that the images were taken in the same room, with identical wall patterns. Using the striped pattern on the wall as a reference, the length of the painting in Image 1 is approximately equal to the width of 14 wall stripes, while the length of the table beneath the mirror in Image 2 is about 12 wall stripes wide. Therefore, the painting in Image 1 is longer than the table beneath the mirror in Image 2.

Figure 22: Full example for Attribute (Measurement) category.

Attribute (Appearance)

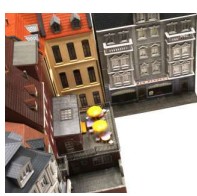 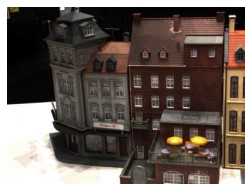

Question: What is the shape of the top view of the visible part of the building in the images?
Options: A: I-shape, B: T-shape, C: F-shape, D: L-shape

Reasoning: The yellow umbrella and the colorful rooftops in Images 1 and 2 have overlapping areas, allowing for spatial reasoning when the two images are combined. In Image 2, four houses are connected to form a vertical rectangle, while in Image 1, there are two small horizontal rectangles on the same side of it. The vertical rectangle forms the main stem of the letter "F", and the two horizontal rectangles form the top and middle bars of the "F". Therefore, the shape of the top view of the visible part of the building is the shape of "F".

Figure 23: Full example for Attribute (Appearance) category.

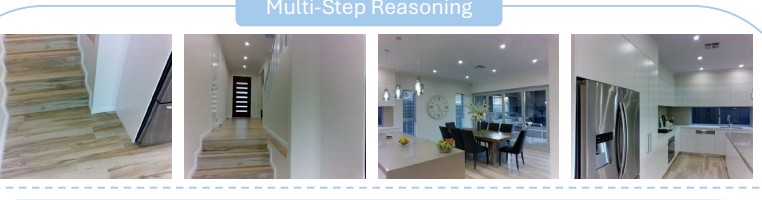

**Multi-Step Reasoning**

Question: When I open the door and come in from outside, where is the clock located relative to me?
Options: A: Directly in front, B: On the left, C: Front right, D: Front left

Reasoning: The staircase appears in both Image 1 and Image 2, the refrigerator in the lower right corner of Image 1 overlaps with the refrigerator on the left side of Image 2, and the tiled tabletop on the left side of Image 3 overlaps with the right side of Image 4. Therefore, the sequence of the images is as follows: from Image 2 to Image 1, the camera turns downward; from Image 1 to Image 4, the camera turns to the right; from Image 4 to Image 3, the camera turns to the right. Based on the overlaps and the order of the images, when I open the door, come in from outside, I am facing perpendicular to the floor-to-ceiling window seen in Image 3. To reach the clock shown in Image 3, I should walk straight and then turn left. Thus, the clock is located in the front-left direction relative to me.

Figure 24: Full example for Multi-Step Reasoning category.

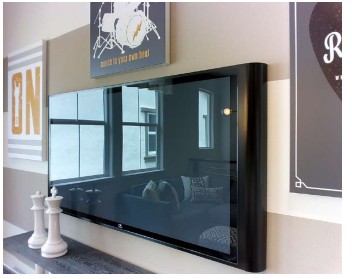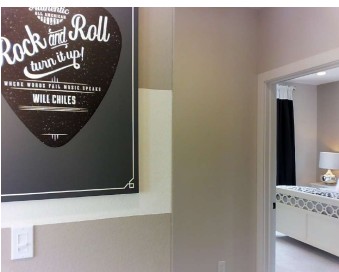

If the direction the TV is facing towards the viewer is considered the front, where is the bedroom located relative to the TV?
Options: A: Front left, B: Back left, C: Front right, D: Back right

**GPT-4o**

To determine the location of the bedroom relative to the TV, let's analyze the images step by step: 1. First Image Analysis: - The TV is mounted on the wall. - There are decorative items and artwork around the TV. - The reflection in the TV shows a window and part of a room with a couch. 2. Second Image Analysis: - The second image shows a view of a bedroom through a doorway. - The artwork on the wall in the second image is the same as the one on the right side of the TV in the first image. 3. Relative Positioning: - Since the artwork is on the right side of the TV in the first image, and the bedroom is visible through a doorway next to this artwork in the second image, the bedroom is located to the right of the TV. 4. Conclusion: - The bedroom is located to the right of the TV when facing it. Since the TV is facing towards the viewer, the bedroom is on the viewer's right side. The bedroom is located at the back right relative to the TV. The correct option is: D.

**GPT-4.5**

Step-by-step reasoning: 1. The TV is facing towards the viewer, so the viewer is looking at the front of the TV. 2. In the first image, we see the reflection on the TV screen, showing windows and a sofa, indicating the room extends behind and to the left of the TV. 3. The second image shows the bedroom through a doorway, which is located to the right side of the TV (as the TV is not visible in this second image, it must be behind the viewer's perspective). 4. Since the bedroom is to the right and behind the viewer's perspective (opposite the front of the TV), the bedroom is located at the back right relative to the TV. Therefore, the correct answer is `D`.

**Qwen2.5-VL-72B**

To determine the location of the bedroom relative to the TV, let's analyze the images step by step: 1. Image 1 : The TV is mounted on the wall, and we can see a reflection of a window and some furniture in the TV screen. This indicates that the TV is facing towards the viewer. 2. Image 2 : The image shows a poster on the wall and an open doorway leading to another room. The doorway is on the right side of the image, and through it, we can see part of a bedroom with a bed and a lamp. 3. Relative Positioning : Since the TV is facing towards the viewer in Image 1, and the bedroom is visible through the doorway on the right side of Image 2, the bedroom must be located to the right of the TV when considering the TV's orientation. Given these observations, the bedroom is located to the front right of the TV. ``C``

Figure 25: Illustration of situation-transformation reasoning errors (highlighted in orange).

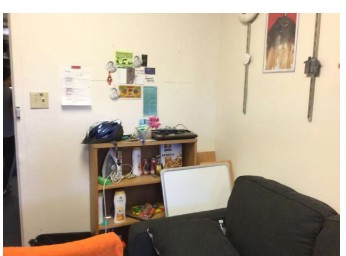 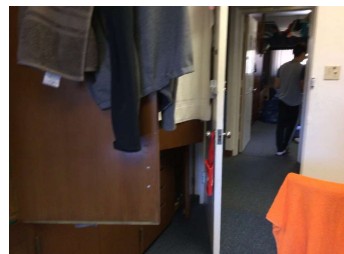

The pictures are taken sequentially from a first-person perspective. At the moment of the last picture, in which direction is the shelf relative to you?
Options: A: back left, B: front right, C: back right, D: front left

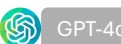 GPT-4o

To determine the direction of the shelf relative to the viewer in the last picture, let's analyze the sequence of images: 1. First Image: - The shelf is visible on the right side of the image. - The couch is in the foreground, suggesting the viewer is facing the shelf. 2. Second Image: - The viewer is looking through a doorway, and the shelf is no longer visible. - The person is walking away from the viewer, indicating the viewer has turned around. Given this sequence, in the last image, the viewer has turned away from the shelf. Therefore, the shelf is now behind the viewer and to the right. The correct option is C: back right.

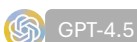 GPT-4.5

Step-by-step reasoning: 1. The first image shows the shelf directly in front of the viewer, slightly to the right. 2. The second image is taken by turning the view to the left, as indicated by the shelf no longer being visible and the door and hallway now being in front. 3. Since the shelf was initially in front and slightly to the right, after turning left, the shelf must now be behind and to the right of the viewer. Therefore, the shelf is located at the back right relative to the viewer. Answer: `C`

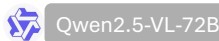 Qwen2.5-VL-72B

To determine the direction of the shelf relative to the person taking the last picture, let's analyze the sequence of images: 1. In the first image, we see a room with a shelf on the left side. The person appears to be facing the shelf. 2. In the second image, the person has moved forward and turned slightly to the right, as indicated by the change in the view of the room and the presence of another person in the background. Given this movement: - The shelf that was initially on the left side of the first image is now behind and to the left of the person in the second image. At the moment of the last picture, the shelf is relative to the person as \"A\".

Figure 26: Illustration of overlap-matching and scene-reconstruction errors (highlighted in red), spatial-logic errors (highlighted in green).

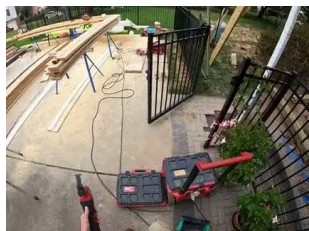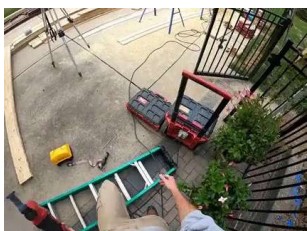

The images are taken sequentially from a first-person perspective. In which direction is the camera moving (with the direction the camera is facing at the moment of image 1 as the front)?
Options: A: Backward, B: Back right, C: Left, D: Back left

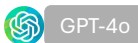 GPT-4o

To determine the direction of movement, let's analyze the changes between the two images: 1. Image 1: - The person is holding a tool and standing near a gate. - There are two toolboxes on the ground, and a ladder is visible to the left. 2. Image 2: - The person is now closer to the ladder, which is more prominently in view. - The toolboxes are still visible, but the perspective has shifted slightly to the left. - The person seems to have moved closer to the ladder and slightly away from the gate. Based on these observations, the movement appears to be towards the ladder and slightly away from the initial position, which suggests a movement to the back left relative to the initial facing direction. Therefore, the direction of movement is D: Back left.

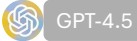 GPT-4.5

Step-by-step reasoning: 1. In the first image, observe the position of the gate, the toolbox, and the plants. The gate is slightly open, and the toolbox is directly in front of the person. 2. In the second image, the gate and toolbox appear to have moved slightly forward and to the right relative to the person's perspective. The plants, which were initially on the right side, are now more directly in front. 3. This indicates that the person has moved backward and slightly to the left relative to their original facing direction in image 1. Therefore, the correct direction of movement is "Back left". Answer: `D`

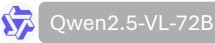 Qwen2.5-VL-72B

To determine the direction of movement, let's analyze the changes between the two images: 1. Image 1: The person is holding a tool and appears to be in a workshop area with various tools and materials around. The perspective suggests they are facing towards the gate and the outdoor area. 2. Image 2: The person is now on a ladder, and the view has shifted. The workshop area is still visible but from a different angle. The person appears to be moving towards the left side of the image, as indicated by the position of the ladder and the change in the surrounding environment. Given these observations, the person seems to be moving to the left relative to their initial position in Image 1. Therefore, the correct answer is: `C`

Figure 27: Illustration of grounding errors (highlighted in blue), spatial-logic errors (highlighted in green).

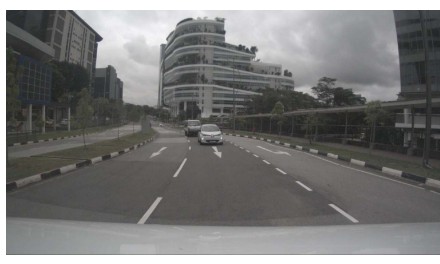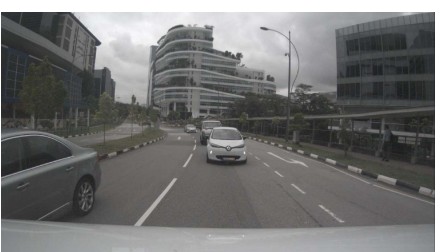

Sequential shooting from a first-person perspective, with the direction the camera is facing as the front, in which direction is the camera moving?
Options: A: Left, B: Forward, C: Backward, D: Right

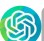 GPT-4o

To determine the direction the camera is moving, let's analyze the two images: 1. First Image: - The white car is in the middle lane, moving towards the camera. - The camera is positioned behind the white car. 2. Second Image: - The white car is closer to the camera, indicating that the camera has moved closer to the car. - The camera is still positioned behind the white car, but the car is now larger in the frame. From these observations, the camera is moving towards the white car, which means it is moving forward. Therefore, the direction the camera is moving is B: Forward.

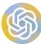 GPT-4.5

Step-by-step reasoning: 1. In the first image, the white car is directly ahead and slightly distant, and the road markings indicate the camera is in the same lane as the white car. 2. In the second image, the white car is closer to the camera, and the silver car on the left side is now visible and closer, indicating the camera has moved backward relative to the white car. 3. The relative positions of the vehicles and road markings clearly show that the camera is moving away from the white car, meaning it is moving backward. Therefore, the correct answer is `C`.

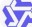 Qwen2.5-VL-72B

To determine the direction the camera is moving, let's analyze the changes between the two images: 1. Position of Vehicles: In the second image, the white car in front appears slightly closer, indicating that the camera (and the vehicle it is attached to) is moving forward. 2. Surrounding Environment: The buildings and trees on both sides of the road appear to be in similar positions relative to the camera, suggesting no significant lateral movement (left or right). 3. Road Markings: The road markings and the overall layout of the road remain consistent, further supporting the idea of forward movement. Given these observations, the camera is moving forward. ``B``

Figure 28: Illustration of spatial-logic errors (highlighted in green).

