# OpenReview forum: "MMSI-Bench: A Benchmark for Multi-Image Spatial Intelligence"
_ICLR.cc/2026/Conference — ICLR 2026 Poster_

### Official Review · Reviewer_FDLo · 2025-10-26

**Soundness:** 4
**Presentation:** 4
**Contribution:** 3
**Rating:** 8
**Confidence:** 4

**Summary:**

This paper introduces MMSI-Bench, a new VQA benchmark for evaluating the multi-image spatial intelligence of MLLMs. The dataset consists of 1,000 challenging, multiple-choice questions meticulously crafted by 3D-vision researchers, which are designed to be unanswerable from any single image. Each question is paired with a human-annotated, step-by-step reasoning process and is categorized into one of eleven spatial reasoning tasks. An extensive evaluation of 37 MLLMs reveals a substantial 55-point performance gap between human (97.2%) and SOTA model (41.9%) accuracy. The authors also provide an automated error analysis pipeline, identifying "overlap-matching and scene-reconstruction" as the dominant failure mode for current models.

**Strengths:**

1. The paper addresses a clear and important gap in MLLM evaluation, moving beyond existing benchmarks that often focus on single-image reasoning or use automated templates.
2. The dataset's fully human-centric curation process, involving 3D-vision experts and 8 diverse, real-world data sources, produces challenging and linguistically varied questions.
3. The authors provide an extensive evaluation of 37 MLLMs, establishing a robust baseline and highlighting a massive performance gap between SOTA models and humans.
4. The inclusion of an automated error analysis pipeline, which categorizes failures into intuitive types, provides concrete and actionable directions for future research.
5. The investigation into prompting techniques, including a novel visual prompting method using feature matching, provides strong evidence that current models have fundamental limitations in this domain.

**Weaknesses:**

1. The dataset size of 1,000 samples, while acknowledged as a result of costly human curation, is small. This raises questions about the in-depth diversity within each of the 10 sub-categories, which have ~100 samples or fewer on average.
2. The "multi-image" claim feels overstated, as all categories except for "Multi-Step Reasoning" are explicitly constrained to using exactly two images.
3. Key results that strongly support the benchmark's novelty, such as the poor performance of models specifically finetuned on other spatial datasets, are relegated to the appendix.
4. The methodology for constructing "Multi-Step Reasoning" questions is underspecified, lacking detail on how annotators were guided to combine the basic task types.

**Questions:**

1. You demonstrate that zero-shot linguistic and visual prompting fails to provide significant gains. Did you experiment with few-shot in-context learning by providing the full human-annotated reasoning chains as exemplars in the prompt?
2. The error analysis in Figure 7 is aggregated across the entire dataset. Could you provide a breakdown of the error type distributions for the most challenging categories, specifically "Multi-Step Reasoning" and "Motion (Camera)"?
3. The "Positional Relationship" category is very broad, covering six distinct sub-types. Do models show significant performance variation across these sub-categories (e.g., is "Cam-Cam" more difficult than "Obj-Obj")?
4. What specific guidelines were given to the 3D-vision researchers for creating the "Multi-Step Reasoning" questions, and how did you ensure these tasks truly required a sequence of the basic spatial skills?

---

> ### Author Response · Authors · 2025-11-23
> **Author Response to Reviewer FDLo (1/2)**
>
> Dear Reviewer FDLo,
>
> Thank you for your constructive and insightful comments. We appreciate your recognition that our paper **“addresses a clear and important gap in MLLM evaluation,”** **“features a dataset with a fully human-centric curation process,”** **“provides an extensive evaluation of 37 MLLMs,”** **“includes an automated error analysis pipeline,”** **“investigates prompting techniques,”** and **“provides strong evidence that current models have fundamental limitations in this domain.”** We address each question below.
>
> > ***Weakness1:*** The in-depth diversity within each of all sub-categories.
>
> **A1:** Thank you for your question. To analyze diversity within each sub-category, we compute diversity metrics for each and then average them. Specifically, we measure: (1) distinct words, i.e., the total number of unique words after lowercasing and tokenization across all questions, reflecting vocabulary richness; (2) unique n-grams, i.e., the number of unique consecutive n-word sequences (here, n = 3 and 5) after tokenization, reflecting phrasal diversity; and (3) syntactic structures, obtained by parsing each sentence into an abstract syntax tree (removing specific words) and counting the number of unique tree patterns, reflecting the diversity of sentence constructions. We compare this average with other spatial-intelligence benchmarks. As shown in the table, the average diversity per sub-category in MMSI-Bench is clearly higher than that of other spatial-intelligence benchmarks.
>
> | Statistic | LEGO-Puzzle | VSI-Bench | ERQA (Multi-Img) | MMSI-Bench |
> | --- | --- | --- | --- | --- |
> | Distinct words | 93.91 | 100.10 | 135.50 | 255.36 |
> | Unique 3-grams | 182.82 | 611.00 | 377.00 | 1015.73 |
> | Unique 5-grams | 201.09 | 1068.50 | 466.25 | 1299.82 |
> | Syntactic structures | 11.90 | 47.40 | 44.38 | 75.09 |
>
>
> > ***Weakness2:*** The "multi-image" claim.
>
> **A2:** Here, “multi-image” is used in contrast to “single-image,” following the notion introduced in works such as MMIU [1], PRIMA [2], and MIRB [3]. Our focus is on spatial understanding with more than one image. As discussed in Appendix C, we aim to provide clearer error diagnosis. Since current models still perform quite poorly, improving performance requires starting from basic capabilities. Therefore, we deliberately concentrate on the simplest multi-image setting, i.e., two images, as the fundamental case. We then further extend to three or more images when constructing the Multi-Step Reasoning category.
>
> [1] Meng, Fanqing, et al. "MMIU: Multimodal Multi-image Understanding for Evaluating Large Vision-Language Models." The Thirteenth International Conference on Learning Representations.
>
> [2] Wahed, Muntasir, et al. "Prima: Multi-image vision-language models for reasoning segmentation." arXiv preprint arXiv:2412.15209 (2024).
>
> [3] Zhao, Bingchen, et al. "Benchmarking multi-image understanding in vision and language models: Perception, knowledge, reasoning, and multi-hop reasoning." arXiv preprint arXiv:2406.12742 (2024).
>
>
> > ***Weakness3:*** Including key results strongly supporting the benchmark's novelty in the main text.
>
> **A3:** Thank you for the suggestion. In the revised version, we have moved these key results from the appendix into the main text (line 358).
>
>
> > ***Weakness4 & Question4:*** Guidelines for constructing "Multi-Step Reasoning" questions and ensuring these tasks require a sequence of basic spatial skills.
>
> **A4:** For the “Multi-Step Reasoning” type, we ask annotators to design questions that are as challenging as possible and cannot be answered directly, and to explicitly write down the step-by-step solution in the reasoning process. According to our statistics on human-annotated reasoning processes in this category, “Multi-Step Reasoning” questions involve between 3 and 11 reasoning steps, with an average of 5.7 steps per question. We have clarified these details in the revised version.

---

> ### Author Response · Authors · 2025-11-23
> **Author Response to Reviewer FDLo (2/2)**
>
> > ***Question1:*** An experiment with few-shot in-context learning by providing the human-annotated reasoning chains in the prompt.
>
> **A5:** Thank you for the feedback. We conducted few-shot in-context learning experiments where we provided the full human-annotated reasoning chains (from questions outside MMSI-Bench) in the prompt and evaluated GPT-4o. We found that few-shot in-context learning also did not lead to a significant performance gain. The results are shown in the table below.
>
>
> | Model  | Overall | Change |
> |--------|:-------:|:------:|
> | GPT-4o | 0.303   |     -   |
> | 1-shot | 0.317   |    +0.014    |
> | 3-shot | 0.288   |    -0.015    |
>
> > ***Question2:*** A breakdown of the error type distributions for the most challenging categories, specifically "Multi-Step Reasoning" and "Motion (Camera)".
>
> **A6:** Thank you for the suggestion. For GPT-4o, the error-type distributions on the “Motion (Camera)” and “Multi-Step Reasoning (MSR)” tasks are as follows:
>
> **GPT-4o – Motion (Camera)**
> - Grounding error: 8 (10.81%)
> - Overlap identification & scene reconstruction error: 25 (46.04%)
> - Situation transformation reasoning error: 1 (1.35%)
> - Spatial logic error (other): 7 (9.46%)
> - Completely correct reasoning: 24 (32.43%)
>
> **GPT-4o – Multi-Step Reasoning (MSR)**
> - Grounding error: 20 (10.10%)
> - Overlap identification & scene reconstruction error: 100 (50.51%)
> - Situation transformation reasoning error: 29 (14.65%)
> - Spatial logic error (other): 3 (1.52%)
> - Completely correct reasoning: 46 (23.24%)
>
> > ***Question3:*** Performance variation across sub-categories of the "Positional Relationship" category.
>
> **A7:** Thank you for the question. We evaluate the accuracies of all models in Table 3 on these sub-categories and compute the average accuracy for each type:
>
> - Obj–Obj: 28.9
> - Cam–Cam: 29.0
> - Cam–Obj: 30.0
> - Reg–Reg: 30.2
> - Obj–Reg: 33.8
> - Cam–Reg: 33.6
>
> From these averages, we can see that the Reg-related sub-categories are generally easier, which may be because regions are larger and therefore easier to localize. By contrast, Obj–Obj is the most difficult on average, possibly because objects tend to be smaller and harder to localize precisely. Cam–Cam is also relatively difficult, likely because some camera movements in these questions are very small and thus require fine-grained visual comparison; in addition, current models’ training data may generally lack rich first-person-view annotations, making it harder for them to understand camera position changes.
>
> > ***Question4:*** Guidelines for constructing "Multi-Step Reasoning" questions and ensuring these tasks require a sequence of basic spatial skills.
>
> The detailed response can be found in ***Weakness4***.
>
> Thank you once again for your time!

---

### Official Review · Reviewer_AFVc · 2025-10-31

**Soundness:** 3
**Presentation:** 3
**Contribution:** 3
**Rating:** 4
**Confidence:** 3

**Summary:**

This paper presents MMSI-Bench, a benchmark designed to evaluate multi-image spatial intelligence in multimodal large language models (MLLMs). The dataset contains 1,000 human-curated, multiple-choice questions requiring reasoning across multiple real-world images, with detailed reasoning annotations. The authors categorize 11 spatial reasoning tasks covering position, motion, and attributes, and benchmark 37 models. Results show a large gap between current MLLMs (best at \~42%) and human performance (\~97%), revealing major weaknesses in spatial reasoning across viewpoints. They also propose an automated error-analysis pipeline that identifies key failure types such as grounding and scene-reconstruction errors.

**Strengths:**

Originality: The focus on multi-image spatial reasoning fills a clear gap between single-image VQA and real-world embodied perception. The fully human-curated design adds credibility compared to prior template-based datasets.

Quality: The taxonomy of spatial relations (camera, object, region) is systematic, and the annotation process with reasoning traces and multi-reviewer verification shows rigor. The large-scale evaluation across 37 models is comprehensive and carefully controlled.

Clarity: The paper is well organized with clear figures and strong examples of question categories. The error typology (grounding, scene reconstruction, situation transformation, spatial logic) provides insight beyond raw accuracy.

Significance: MMSI-Bench exposes a real bottleneck in MLLMs’ ability to perform grounded spatial reasoning. The benchmark can drive future work on embodied AI, robotics, and multi-view understanding.

**Weaknesses:**

The dataset is still modest in scale (1k QA pairs), which limits generalization analysis. It would help to report variability or cross-split reliability. Many questions rely on human interpretation of viewpoint or direction. Some ambiguity might remain even with expert curation, which could affect reproducibility.

The evaluation metric focuses only on answer accuracy; assessing reasoning trace similarity (e.g., using annotated rationales) could reveal finer-grained improvements. While the benchmark is thorough, the paper lacks concrete recommendations or model design principles derived from the findings.

**Questions:**

1. How was question difficulty calibrated beyond human answer time? Did annotators estimate complexity or confidence?

2. Could reasoning annotations be used for training models (not just analysis)? If so, how does that affect overfitting?

3. How consistent are human annotators across the four error types? Any quantitative measure?

4. Did you observe differences between models trained with ego-centric data vs general web-image pretraining?

5. How might MMSI-Bench interact with embodied datasets like Habitat or RoboBrain for active perception tasks?

---

> ### Author Response · Authors · 2025-11-23
> **Author Response to Reviewer AFVc (1/4)**
>
> Dear Reviewer AFVc,
>
> Thank you for your constructive and insightful comments. We appreciate your recognition of our work’s **“Originality,”** **“Quality,”** **“Clarity,”** and **“Significance.”** We address each question below.
>
> > ***Weakness1.1:*** Report variability or cross-split reliability.
>
> **A1.1:** Thank you for this suggestion. We conduct a repeated random sub-sampling evaluation to quantify variability. Specifically, on the 1k QA pairs of MMSI-Bench, we randomly sample subsets of size 20, 40, 60, 80, 100, 200, 300, 500, 700, and 1000, and evaluate GPT-4o on each subset. For each subset size, we perform 3 independent samplings and compute the standard deviation as well as the variance of accuracy across the 3 runs. The results are shown in the following table.
>
> From these results, we observe that when the sample size is around 60 (which is already clearly smaller than the number of samples per task), the standard deviation and variance of accuracy are already within a reasonably small and acceptable range, and the mean accuracy also begins to stabilize once the sample size exceeds 100. As the sample size increases beyond 500, the standard deviation and variance of accuracy become very small, indicating that the overall accuracy estimates are stable. Therefore, we believe that the experimental results on MMSI-Bench are reliable.
>
> | Sample Size | Mean Accuracy (%) | Std. Accuracy (%) | Variance of Accuracy |
> |-------------|------------------|-------------------|---------------------|
> | 20          | 31.67            | 4.71              | 0.002222            |
> | 40          | 25.83            | 2.36              | 0.000556            |
> | 60          | 31.33            | 1.36              | 0.000185            |
> | 80          | 28.08            | 1.56              | 0.000243            |
> | 100         | 31.00            | 0.47              | 0.000022            |
> | 200         | 30.00            | 1.08              | 0.000117            |
> | 300         | 29.11            | 1.10              | 0.000121            |
> | 500         | 30.67            | 0.66              | 0.000044            |
> | 700         | 30.52            | 0.47              | 0.000022            |
> | 1000        | 30.30            | 0.00              | 0.000000            |
>
> Finally, we would like to emphasize that MMSI-Bench, with its 1,000 carefully curated samples, is comparable in size to several widely used and high-quality VQA benchmarks, such as MM-Star [1] (1,500 samples), MM-Vet [2] (218 samples), and LLaVA-in-the-Wild [3] (60 samples). While we agree that a larger dataset scale would always be beneficial, we believe that the current size of MMSI-Bench is adequate and its evaluation results are trustworthy.
>
> We have incorporated this discussion into the Limitations section of the updated paper.
>
> [1] Chen, Lin, et al. "Are we on the right way for evaluating large vision-language models?." Advances in Neural Information Processing Systems 37 (2024): 27056-27087.
>
> [2] Yu, Weihao, et al. "MM-Vet: Evaluating Large Multimodal Models for Integrated Capabilities." Forty-first International Conference on Machine Learning.
>
> [3] Liu, Haotian, et al. "Visual instruction tuning." Advances in neural information processing systems 36 (2023): 34892-34916.
>
>
> > ***Weakness1.2:*** Ambiguity of viewpoint or direction.
>
> **A1.2:** Thank you for the question. All of our annotators receive dedicated training to ensure that the information required to answer each question is clearly specified and free of ambiguity. In particular, we explicitly instruct annotators that any question relying on a specific viewpoint or direction must clearly define that viewpoint or direction. In addition, three independent reviewers, separate from the original annotators, systematically examine all data and remove any instances containing ambiguity, to ensure that all questions are clear and answerable.
>
> Moreover, manually constructed and manually verified datasets are typically more reliable than those produced fully automatically. For example, ICLR 2026 Submission 16028 points out that a popular dataset produced by a current automatic data generation pipeline has only a 57.6% human validation rate.

---

> ### Author Response · Authors · 2025-11-23
> **Author Response to Reviewer AFVc (2/4)**
>
> > ***Weakness2.1:*** Assess reasoning trace similarity.
>
> **A2.1:** Thank you for this suggestion. Since the human-annotated reasoning process is not the only valid reasoning path, assessing reasoning trace similarity using human-annotated reasoning processes may not be reliable. In a similar spirit, our paper already provides richer evaluation metrics beyond Multiple-Choice Accuracy by evaluating reasoning correctness, as presented in Figure 7. Specifically, the results for multiple-choice accuracy and reasoning correctness are as follows:
>
> | Model          | Multiple-Choice Accuracy | Reasoning Correctness |
> | :------------- | :----------------------: | :-------------------: |
> | GPT-4.5        |          40.3%           |         37.5%         |
> | GPT-4o         |          30.3%           |         29.9%         |
> | Qwen2.5-VL-72B |          30.7%           |         21.5%         |
>
> We have added a textual elaboration on these results in line 485 in the revised version of the paper to make this point clearer.
>
> Furthermore, our automated analysis framework is capable of analyzing the specific types and proportions of spatial reasoning errors made by various models. This reveals finer-grained insights into their improvements and weaknesses: even when the final answer is correct, the reasoning process can still contain significant errors; and among the various error types, overlap-matching and scene-reconstruction errors account for the largest proportion of mistakes across all models. This is a feature that is lacking in other spatial intelligence benchmarks.
>
>
> > ***Weakness2.2:*** Add concrete recommendations or model design principles derived from the findings.
>
> **A2.2:** Thank you for the suggestion. As most models are not fully transparent about their details, such as training data and model architecture, we cannot provide a very detailed analysis of these aspects. Still, we try to summarize some of our observed insights for model design as follows:
>
> 1. We observe that NVILA-15B performs very well on MMSI-Bench, even surpassing many models with more than 70B parameters. NVILA’s Dynamic-S² technique supports multi-scale high-resolution image processing and can adapt to varying image aspect ratios, avoiding distortion and preserving spatial information. Similarly, the strong open-source model Qwen2.5-VL-72B uses a ViT with dynamic resolution support, which also helps preserve spatial information as much as possible. Therefore, image encoding techniques that adapt to image aspect ratios and resolution are potentially beneficial for spatial understanding.
>
> 2. The training data of NVILA-15B includes CLEVR [1], which contains a large number of spatial-reasoning-related questions. This suggests that including spatially rich data in the training data is helpful for improving spatial intelligence.
>
> 3. We observe that the unified generation–understanding model Bagle [2] achieves strong performance on MMSI-Bench: with only 7B activated parameters, it reaches 31.5% accuracy. This may be because the reconstruction task used during Bagle’s training benefits spatial understanding. Thus, incorporating spatial reconstruction tasks into training is potentially helpful for improving a model’s spatial reasoning ability.
>
> [1] Johnson, Justin, et al. "Clevr: A diagnostic dataset for compositional language and elementary visual reasoning." Proceedings of the IEEE conference on computer vision and pattern recognition. 2017.
>
> [2] Deng, Chaorui, et al. "Emerging properties in unified multimodal pretraining." arXiv preprint arXiv:2505.14683 (2025).

---

> ### Author Response · Authors · 2025-11-23
> **Author Response to Reviewer AFVc (3/4)**
>
> > ***Question1:*** How was question difficulty calibrated beyond human answer time? Did annotators estimate complexity or confidence?
>
> **A3:** Thank you for the question. We explicitly ask annotators to think as carefully and comprehensively as possible to ensure that their answers are correct. For questions that any human solver answered incorrectly, we directly categorize them as the highest difficulty level.
>
> When we use the normalized response time to estimate difficulty, the time already implicitly reflects both the complexity of the question and the solver’s confidence.
>
>
> > ***Question2:*** Train with reasoning annotations.
>
> **A4:** Thank you for this insightful suggestion. We randomly sample 70% of MMSI-Bench for LoRA fine-tuning InternVL2-8B, and use the remaining 30% for testing. The results are shown in the table below.
>
> We observe that fine-tuning using only the correct answer options is hard to learn from and even harms spatial reasoning ability. In contrast, fine-tuning with human-annotated reasoning traces is somewhat helpful, but since the amount of such data is limited, the performance gain is limited and we do not observe a substantial drop in test set performance that would indicate overfitting.
>
> | Model                            | Overall | Change |
> |----------------------------------|:-------:|:------:|
> | InternVL2-8B                     |  26.7   |   -    |
> | FT on correct options only       |  21.7   |   -5.0    |
> | FT on human reasoning annotations|  27.2   |    +0.5   |
>
>
>
>
> > ***Question3:*** Consistency among human annotators for the four error types.
>
> **A5:** In practice, we explicitly ask annotators to label only the single most critical error that directly leads to the subsequent reasoning failure. In our initial annotation pass on 100 samples, for efficiency, each question was checked by a single annotator. To measure consistency, we then employed another annotator to re-check the same set, and there were 4 inconsistent samples across annotators. The annotators subsequently discussed these 4 cases and finally reached consensus on all of them.
>
> > ***Question4:*** Differences between models trained with ego-centric data vs general web-image pretraining.
>
> **A6:** Thank you for your insightful question. Most models do not provide detailed training data recipes, and among the models we test, none explicitly state that they are trained with ego-centric data. We only observe that: InternVL3 is trained with 3D scene understanding data; Seed-1.5-VL is trained with 3D grounding data; and NVILA uses CLEVR, which contains a large number of spatial-reasoning-related questions. Among these models, only NVILA performs well on MMSI-Bench.
>
> Thus, to further improve the spatial reasoning ability of MLLMs, we still face several open problems: how to make effective use of different types of spatial-reasoning-related training data, how to balance their proportions, and how to ensure data quality and diversity.

---

> ### Author Response · Authors · 2025-11-23
> **Author Response to Reviewer AFVc (4/4)**
>
> > ***Question5:*** The correlation between MMSI-Bench and datasets for active perception tasks.
>
> **A7:** Thank you for your valuable feedback. To explore the correlation between MMSI-Bench and datasets for active perception tasks, we conducted an analysis to evaluate the relationship between model performance on our MMSI-Bench and two active perception tasks from EMBODIEDBENCH [1]: EB-Navigation (designed to assess embodied agents’ navigation abilities, where models need to output low-level actions) and EB-Habitat (which leverages the Habitat 2.0 simulator and focuses on planning and executing high-level skills to achieve user-defined goals; it requires robots to visit multiple locations to find desired items).
>
> Our analysis reveals a strong positive correlation between performance on MMSI-Bench and these active perception tasks. Specifically:
>
> - The correlation between MMSI-Bench and EB-Navigation is a Pearson coefficient of 0.8147 and a Spearman coefficient of 0.7333.
>
> - The correlation between MMSI-Bench and EB-Habitat is a Pearson coefficient of 0.7299 and a Spearman coefficient of 0.6167.
>
> These results indicate that a model’s performance on MMSI-Bench is highly correlated with its capabilities in both downstream active perception tasks. The detailed results that form the basis of this analysis are presented in the table below:
>
> | Model                    | MMSI-Bench | EB-Navigation | EB-Habitat |
> |--------------------------|-----------:|----------------------:|------------------------:|
> | GPT-4o                   |    30.3        | 57.7                  | 59.0                    |
> | Claude-3.7-Sonnet        |      30.2      | 45.0                  | 58.7                    |
> | Claude-3.5-Sonnet        |     31.3       | 44.7                  | 68.0                    |
> | Llama-3.2-11B-Vision-Ins |       25.4     | 21.4                  | 25.0                     |
> | InternVL2.5-78B          |      28.5      | 30.7                  | 49.0                    |
> | InternVL2.5-38B          |       27.9     | 30.3                  | 38.3                    |
> | InternVL2.5-8B           |      28.7      | 21.3                  | 11.3                     |
> | Qwen2.5-VL-72B-Ins       |       30.7     | 40.0                  | 37.7                    |
> | Qwen2.5-VL-7B-Ins        |      25.9      | 20.0                  | 14.3                     |
>
> We have incorporated this insightful content into Appendix E.5 of the updated paper. Thank you once again for your valuable advice.
>
> [1] Yang, Rui, et al. "EmbodiedBench: Comprehensive Benchmarking Multi-modal Large Language Models for Vision-Driven Embodied Agents." Forty-second International Conference on Machine Learning.
>
> Thank you once again for your time!

---

### Official Review · Reviewer_HMkb · 2025-10-31

**Soundness:** 3
**Presentation:** 4
**Contribution:** 3
**Rating:** 8
**Confidence:** 4

**Summary:**

The paper introduces MMSI-Bench, a benchmark designed to evaluate the multi-image spatial reasoning capabilities of multimodal large language models (MLLMs). The dataset was created manually by six 3D-vision researchers. It consists of 1,000 multiple-choice question-answer pairs and 1,990 unique images sourced from eight real-world datasets (Matterport3D, ScanNet, DTU, nuScenes, Waymo, AgiBot-World, DAVIS 2017, and Ego4D). The questions are categorized into 11 tasks, under 4 main categories - positional relationships, attribute, motion and multi-step reasoning. The questions revolve around 3 spatial elements - camera, objects, and region. Most questions, other than the multi-step reasoning questions, involve 2 input images. Another key contribution is that each question is accompanied by a human-authored, step-by-step reasoning chain.

The paper conducts a comprehensive evaluation of 37 MLLMs. The primary result is a massive performance gap: the best-performing model (GPT-5) achieves only 41.9% accuracy, while human-level performance is 97.2%. The results also show that multi-step reasoning is particularly challenging for models. The authors also report fine-tuning and prompting ablations both of which provide minimal to no benefit. Language prompting involved zero-shot chain-of-thought reasoning and visual prompting involves highlighting PATS correspondences between image pairs. Finally, the authors provide insights into the failure modes of the evaluated models.

**Strengths:**

1. The benchmark's core strength is its manual, expert-driven annotation process. The questions are linguistically diverse, non-trivial, and require spatial understanding. The problem of multi-image spatial reasoning is highly relevant and timely for advancing embodied AI and robotics, and this paper clearly demonstrates a critical capability gap.
2. The paper evaluates an extensive suite of 37 models, providing a valuable and comprehensive snapshot of the entire SOTA. The inclusion of "Human Performance" (97.2%) is a strong baseline that effectively contextualizes the low model scores.
3. The findings that both advanced prompting (CoT, visual prompting) and fine-tuning fail to provide significant gains are valuable. These negative results strongly suggest that the models' failure is not a simple problem but a more fundamental capability deficit.
4. The paper is well-written making it easy to read and follow.

**Weaknesses:**

1. 1,000 samples is small size, especially when divided across 11 tasks. This limited scale is a direct trade-off for the high-quality manual annotation (300+ hours), but it makes the benchmark difficult to scale and creates a risk of models eventually overfitting to this specific test set.
2. Blind GPT-4o is not a suitable baseline since the questions depend heavily on the images. Language priors are unable to capture the context of the problem unless the images are described in words and the accuracy is expected to be similar to random baseline. The authors should update the baseline to be something more suitable.
3. The paper's analysis of the core problem is unclear. In model size ablations, the authors suggest the bottleneck lies in data quality and diversity. However, the paper's own experiments fine-tuning and prompting show that existing methods to bridge data gaps do not work. This suggests the bottleneck is more likely architectural or requires in-domain fine-tuning, which the paper does not explore.

**Questions:**

1. How are the multiple images fed into the models? Are they concatenated into a single image, as not all models natively support multiple image inputs?
2. The order in which the images are presented to the model seems critical, especially for tasks involving motion. How is this temporal order preserved and communicated to the model during evaluation? Can the authors provide an ablation study on the effect of image ordering?
3. Is the visual prompt (with correspondence lines) provided along with the normal image inputs? There is a concern that the lines themselves might occlude important details in the images, making them harder to see. That could be the reason behind marginal prompting gains.
4. The authors should provide details about the distribution of data across the 11 tasks.

---

> ### Author Response · Authors · 2025-11-23
> **Author Response to Reviewer HMkb (1/2)**
>
> Dear Reviewer HMkb,
>
> Thank you for your constructive and insightful comments. We appreciate your recognition of our benchmark’s **“manual, expert-driven annotation process,”** **“linguistically diverse, non-trivial”** questions, the **“valuable and comprehensive snapshot of the entire SOTA,”** the **“valuable findings,”** and that the paper is **“well-written making it easy to read and follow.”** We address each question below.
>
> > ***Weakness1:*** Scalability and overfitting concerns.
>
> **A1:** Thank you for your insightful comment on scalability. We acknowledge that manual construction is indeed less scalable than fully automated pipelines. For test sets, we believe that data quality is paramount, which is why we opted for manual curation. However, this approach is still potentially scalable by hiring more annotators and providing them with appropriate training.
>
> To provide a concrete cost analysis: with data annotator wages ranging from 1–10 USD per hour, we used an average of 5 USD per hour for our calculations. The 300 annotation hours for MMSI-Bench's 1,000 samples amounted to 1,500 USD. Extrapolating this, expanding the dataset to 10,000 samples—a scale on the order of SFT training data—would cost approximately 15,000 USD. This is roughly equivalent to the rental cost of an 8-GPU H100 server for 25 days, a figure we believe is acceptable and potentially scalable for industry‑level VLM training, because VLM training requires substantial resources. For example, training a state-of-the-art 7B VLM [1] for one time can take up to 400 GPU days, let alone even larger models.
>
> Furthermore, because the questions in MMSI-Bench are not generated from templates, and the images are sourced without a discernible pattern or regularity, it is difficult to manually find a distribution and thus difficult to overfit as discussed in Appendix E.4 of the original paper (line 320 in the updated paper). If a model can show strong performance improvement on our benchmark, we believe it has already achieved a sufficiently strong generalization ability.
>
> We have incorporated this discussion into the Limitations section.
>
> [1] Li B, Zhang Y, Guo D, et al. LLaVA-OneVision: Easy Visual Task Transfer[J]. Transactions on Machine Learning Research.
>
>
>
> > ***Weakness2:*** Blind GPT-4o as baseline.
>
> **A2:** Thank you for your thoughtful quesiton. We would like to clarify that the Blind GPT-4o baseline was not intended for direct performance comparison with other models. Its purpose, as we elaborate in the original paper (line 296), was to demonstrate that our benchmark is highly reliant on visual information and cannot be solved correctly using only question priors or commonsense knowledge.
>
>
>
> > ***Weakness3:*** In model size ablations, the authors suggest the bottleneck lies in data quality and diversity. However, the paper's own experiments fine-tuning and prompting show that existing methods to bridge data gaps do not work. This suggests the bottleneck is more likely architectural or requires in-domain fine-tuning, which the paper does not explore.
>
> **A3:** Thank you for your comment. Our model size ablation shows that within the same architecture family, scaling the model from 7B to larger sizes brings only limited gains on MMSI-Bench. This suggests that simply increasing model size is not an effective way to bridge the current performance gap, so data quality and diversity should be a more critical bottleneck than further scaling model size.
>
> The fine-tuning and prompting experiments in the paper do not contradict this conclusion. Those approaches rely heavily on automatically generated training data, much of which is created from templates. This makes it difficult to guarantee high data quality and leads to insufficient diversity. As a result, the model fails to generalize to the much more diverse data in MMSI-Bench, as we discussed in Appendix E.4 of the original paper (line 320 in the updated paper).
>
> We agree that a better architecture or in-domain fine-tuning may improve performance. Our claim is that, for improving the performance of a small model such as 7B, prioritizing improvements in data over simply scaling up the model size is more efficient.
>
> Very recent work independently supports the central role of data. For example, SenseNova-SI [1] and Visual Spatial Tuning [2] carefully construct high-quality, diverse spatial reasoning data and fine-tune relatively small models. Their results show substantial performance improvements on multiple benchmarks, including MMSI-Bench, in some cases surpassing much larger models.
>
> [1] Cai, Zhongang, et al. "Scaling Spatial Intelligence with Multimodal Foundation Models." arXiv preprint arXiv:2511.13719 (2025).
>
> [2] Yang, Rui, et al. "Visual Spatial Tuning." arXiv preprint arXiv:2511.05491 (2025).

---

> ### Author Response · Authors · 2025-11-23
> **Author Response to Reviewer HMkb (2/2)**
>
> > ***Question1:*** Multi-image input method and support.
>
> **A4:** Each image is input separately; there is no concatenation. All the models we tested, as well as the current mainstream MLLMs, natively support multiple image inputs.
>
> > ***Question2:*** Temporal order preservation method and effect of image order.
>
> **A5:** Thank you for the question. For cases involving temporal order, we explicitly describe the temporal relationships among the images in the question. The model can perceive the images in a specific order due to the positional encoding of the LLM.
>
> Following your question, we performed an ablation study on the Motion-Cam subcategory of MMSI-Bench (in which answering every question depends on the temporal order of images) by reversing the image order and testing GPT-4o. The performance decreased significantly, as shown below.
>
> | Setting                             | GPT-4o Performance | Change    |
> | ----------------------------------- | ------------------ | --------- |
> | Motion-Cam. (Original)                | 35.1%              |      -     |
> | Motion-Cam. (Reversed Image Order)   | 16.2%              | -18.9%   |
>
>
>
> > ***Question3:*** Visual prompt provision method and concern of occluding image details.
>
> **A6:** Thank you for this insightful question. To confirm, yes, the visual prompt is provided along with the normal image inputs.
>
> To minimize any potential occlusion, we conducted experiments where we modified the visual prompts to use significantly fewer and thinner lines. However, our results showed that these less obtrusive visual prompts did not lead to any performance improvement. Therefore, we conclude that occlusion from the correspondence lines is not the reason for the marginal gains observed with our visual prompting method. The detailed results are as follows:
>
> | Setting                                   | GPT-4o Performance | Change  |
> |--------------------------------------------|--------------------|---------|
> | MMSI-Bench (no visual prompt)              | 30.3%              | –       |
> | + Visual prompt (number: 16, width: 3)<br>&nbsp;&nbsp;&nbsp;(setting from the original paper)          | 31.1%              | +0.8%   |
> | + Visual prompt (number: 16, width: 2)          | 30.4%              | +0.1%   |
> | + Visual prompt (number: 16, width: 1)          | 29.7%              | –0.6%   |
> | + Visual prompt (number: 8, width: 1)           | 30.6%              | +0.3%   |
> | + Visual prompt (number: 2, width: 1)           | 29.8%              | –0.5%   |
>
> > ***Question4:*** Details about the distribution of data across the 11 tasks.
>
> **A7:** Thank you for your suggestion. We have provided details about the distribution of data across the 11 tasks in the appendix in the revised version.
>
> Thank you once again for your time!

---

### Official Review · Reviewer_htiq · 2025-11-01

**Soundness:** 3
**Presentation:** 2
**Contribution:** 3
**Rating:** 6
**Confidence:** 4

**Summary:**

This paper introduces MMSI-Bench, a large-scale benchmark for evaluating multi-image spatial intelligence in multimodal large language models (MLLMs).
Unlike prior benchmarks focusing on single-image reasoning, MMSI-Bench evaluates an MLLM’s ability to reason across multiple images to infer spatial relationships, motion, and object-camera-region dynamics.
The dataset contains 1,000 multiple-choice questions (covering 10 atomic spatial reasoning categories and one multi-step reasoning task) curated by experts from over 120,000 real-world images drawn from datasets such as ScanNet, nuScenes, Matterport3D, and Ego4D. Each question includes human-written reasoning chains and carefully designed distractors.
Extensive evaluations of 37 MLLMs (including GPT-5, Gemini-2.5, Claude-3.7, Qwen2.5-VL, and InternVL-3) reveal that even the best proprietary model achieves only ~42% accuracy, while humans reach 97%, showing a large performance gap. The paper also presents an automated error analysis framework identifying four dominant failure modes: grounding, overlap-matching, situation-transformation, and spatial-logic errors.

**Strengths:**

Novel Benchmark Scope: MMSI-Bench uniquely targets multi-image spatial reasoning — a critical yet underexplored capability for MLLMs and embodied AI systems. Prior works (e.g., BLINK, ReMI, MuirBench) only contain limited spatial sub-splits, while this benchmark provides systematic coverage.

High-Quality, Human-Curated Data: Each question is manually designed and audited by multiple experts with reasoning explanations, ensuring clarity, difficulty, and lack of ambiguity. The benchmark’s construction pipeline (Fig. 4) and taxonomy (Table 1) are well-documented and rigorous.

Comprehensive Evaluation: The authors benchmark 37 models, analyze scaling trends, compare open-source and proprietary systems, and examine effects of CoT and visual prompting. This breadth enhances credibility.

Insightful Error Taxonomy: The four-type categorization of reasoning errors (Fig. 6) — grounding, scene reconstruction, situation transformation, and spatial logic — provides clear direction for future model development.

Impactful Findings: The results demonstrate that current MLLMs lack robust spatial reasoning and that scaling model size or prompt engineering yields marginal gains, implying fundamental architectural and data limitations.

Strong Writing and Presentation: Figures and examples (e.g., Fig. 2’s diverse question types) are clear, and the organization is consistent with ICLR standards.

**Weaknesses:**

Manual Effort vs. Scalability: Although the manual curation ensures quality, it also limits scalability — future expansions may face bottlenecks unless semi-automatic generation or verification methods are introduced.

Metric Simplicity: The benchmark reports only accuracy on multiple-choice tasks. Incorporating richer evaluation metrics (e.g., reasoning correctness or step alignment) could offer more granular insight.

Potential Dataset Bias: While data diversity is claimed, the benchmark draws primarily from common 3D and driving datasets, possibly biasing toward indoor and urban scenes rather than outdoor natural environments.

Limited Generalization Discussion: The paper does not test transfer to downstream embodied tasks (e.g., navigation, manipulation) where multi-view reasoning is crucial.

**Questions:**

See Weaknesses

---

> ### Author Response · Authors · 2025-11-23
> **Author Response to Reviewer htiq (1/2)**
>
> Dear Reviewer htiq,
>
> Thank you for your constructive and insightful comments. We appreciate your recognition of our work’s **“Novel Benchmark Scope,”** **“High-Quality, Human-Curated Data,”** **“Comprehensive Evaluation,”** **“Insightful Error Taxonomy,”** **“Impactful Findings,”** and **“Strong Writing and Presentation.”** We address each question below.
>
> > ***Weakness1:*** Manual Effort vs. Scalability.
>
> **A1:** Thank you for your insightful comment on scalability. We acknowledge that manual construction is indeed less scalable than fully automated pipelines. For test sets, we believe that data quality is paramount, which is why we opted for manual curation. However, this approach is still potentially scalable by hiring more annotators and providing them with appropriate training.
>
> To provide a concrete cost analysis: with data annotator wages ranging from 1–10 USD per hour, we used an average of 5 USD per hour for our calculations. The 300 annotation hours for MMSI-Bench's 1,000 samples amounted to 1,500 USD. Extrapolating this, expanding the dataset to 10,000 samples—a scale on the order of SFT training data—would cost approximately 15,000 USD. This is roughly equivalent to the rental cost of an 8-GPU H100 server for 25 days, a figure we believe is acceptable and potentially scalable for industry‑level VLM training, because VLM training requires substantial resources. For example, training a state-of-the-art 7B VLM [1] for one time can take up to 400 GPU days, let alone even larger models.
>
> You are right that semi-automatic generation or verification methods are very important. This remains a challenging open question, and we will further explore this direction in future work.
>
> We have incorporated this discussion into the Limitations section of the updated paper.
>
> [1] Li B, Zhang Y, Guo D, et al. LLaVA-OneVision: Easy Visual Task Transfer[J]. Transactions on Machine Learning Research.
>
>
> > ***Weakness2:*** Need for richer evaluation metrics.
>
> **A2:** Thank you for this suggestion. In addition to the multiple-choice accuracy, our paper has evaluated the reasoning correctness, which is presented in Figure 7. Specifically, the results for multiple-choice accuracy and reasoning correctness are as follows:
>
> | Model          | Multiple-Choice Accuracy | Reasoning Correctness |
> | :------------- | :----------------------: | :-------------------: |
> | GPT-4.5        |          40.3%           |         37.5%         |
> | GPT-4o         |          30.3%           |         29.9%         |
> | Qwen2.5-VL-72B |          30.7%           |         21.5%         |
>
> We have added a textual elaboration on these results in line 485 in the revised version of the paper to make this point clearer.
>
> Furthermore, our automated analysis framework is capable of analyzing the specific types and proportions of spatial reasoning errors made by various models. This reveals finer-grained insights into their improvements and weaknesses, which is a feature that is lacking in other spatial intelligence benchmarks.

---

> ### Author Response · Authors · 2025-11-23
> **Author Response to Reviewer htiq (2/2)**
>
> > ***Weakness3:*** Possible bias toward indoor/urban scenes rather than outdoor natural environments.
>
> **A3:** Thank you for your insightful comment regarding potential dataset bias. In fact, our data sources such as Ego4D and DAVIS 2017 contain outdoor natural environments. Additionally, our 3D datasets include recordings in yards, and side-facing cameras in our driving datasets capture rural scenes.
>
> We have conducted further experiments to demonstrate that the benchmark is not biased against outdoor natural environments. Specifically, we identified the questions in MMSI-Bench involving outdoor natural environments and tested model performance on this subset. The results are similar to the overall performance on MMSI-Bench, and we did not observe a significant bias. The results are as follows:
>
> | Model                | Overall | Outdoor Natural Environments |
> |----------------------|--------------------|-----------------------------------------|
> | GPT-4.5              |        40.3            |       38.1                                  |
> | GPT-4o               |       30.3             |       29.9                                  |
> | Gemini-2.5-Pro       |         36.9           |        37.1                                 |
> | Qwen2.5-VL-72B       |         30.7           |       27.8                                  |
> | LLaVA-OneVision-72B  |     28.4               |        30.9                                 |
> | LLaVA-OneVision-7B   |        24.5            |        26.8                                 |
>
>
>
>
> > ***Weakness4:*** Generalization to downstream embodied tasks (e.g., navigation, manipulation).
>
> **A4:** Thank you for your valuable feedback. We have conducted an analysis to evaluate the correlation between model performance on our MMSI-Bench and two critical downstream tasks from EMBODIEDBENCH [1]: EB-Navigation and EB-Manipulation.
>
> Our analysis reveals a strong positive correlation between performance on MMSI-Bench and these embodied tasks. Specifically:
>
> - The correlation between MMSI-Bench and EB-Navigation is a Pearson coefficient of 0.8147 and a Spearman coefficient of 0.7333.
> - The correlation between MMSI-Bench and EB-Manipulation is a Pearson coefficient of 0.7402 and a Spearman coefficient of 0.7500.
>
> These results indicate that a model's performance on MMSI-Bench is highly correlated with its capabilities in both downstream navigation and manipulation tasks. Notably, the correlation with navigation is particularly high, which suggests that the spatial understanding abilities evaluated by MMSI-Bench are closely aligned with the skills required for navigation tasks. The detailed results that form the basis of this analysis are presented in the table below:
>
> | Model                    | MMSI-Bench | EB-Navigation | EB-Manipulation |
> |--------------------------|-----------:|----------------------:|------------------------:|
> | GPT-4o                   |    30.3        | 57.7                  | 39.6                    |
> | Claude-3.7-Sonnet        |      30.2      | 45.0                  | 28.5                    |
> | Claude-3.5-Sonnet        |     31.3       | 44.7                  | 25.4                    |
> | Llama-3.2-11B-Vision-Ins |       25.4     | 21.4                  | 0.9                     |
> | InternVL2.5-78B          |      28.5      | 30.7                  | 10.6                    |
> | InternVL2.5-38B          |       27.9     | 30.3                  | 15.8                    |
> | InternVL2.5-8B           |      28.7      | 21.3                  | 7.0                     |
> | Qwen2.5-VL-72B-Ins       |       30.7     | 40.0                  | 16.2                    |
> | Qwen2.5-VL-7B-Ins        |      25.9      | 20.0                  | 9.6                     |
>
> We have incorporated this insightful content into Appendix E.5 of the updated paper. Thank you once again for your valuable advice.
>
> [1] Yang, Rui, et al. "EmbodiedBench: Comprehensive Benchmarking Multi-modal Large Language Models for Vision-Driven Embodied Agents." Forty-second International Conference on Machine Learning.
>
> Thank you once again for your time!

---

### Author Response · Authors · 2025-11-23
**General Response**

Dear Reviewers, AC, SAC, and PC,

Thank you for your efforts in organizing ICLR and coordinating the review process. We sincerely thank all reviewers for their insightful comments and constructive feedback.

We are grateful that our work was recognized for its **originality and benchmark scope**, **the fully human, expert-driven curation process**, **the linguistic diversity and non-triviality of the questions**, **the comprehensive evaluation**, **the automated and fine-grained error analysis**, **the impactful empirical findings** (especially on current models’ spatial limitations), and **the clarity and organization of the paper**.

Our responses focused on several core concerns:

1. **Scalability of a manually constructed benchmark**
   We show that MMSI-Bench is comparable in scale to existing spatial benchmarks, and its fully human-curated design can be further expanded at acceptable cost if needed. At the same time, current models still exhibit substantial headroom on our tasks, indicating that the present scale of the benchmark is sufficient.

2. **Richer evaluation metrics**
   We addressed this by reporting reasoning correctness and detailed error taxonomies in addition to accuracy.

3. **Potential biases in data distribution** (e.g., indoor/urban vs. outdoor natural scenes)
   Targeted subset experiments showed no bias.

4. **Relevance to downstream embodied and active perception tasks**
   We demonstrated strong correlations with multiple EmbodiedBench tasks, supporting the practical value of MMSI-Bench.

5. **Implementation and design details**
   We clarified and strengthened aspects such as multi-image input, temporal order, visual prompting, difficulty calibration, annotator consistency, and intra-category diversity, supported by additional ablations and statistics.

These points have been incorporated into the revised version to make the benchmark and findings more robust, transparent, and useful to the community.

We sincerely appreciate the reviewers’ constructive suggestions and remain committed to further improving our work.

Below, we address each reviewer’s specific concerns in detail. We extend our heartfelt thanks to all reviewers for their recognition and valuable feedback. We welcome further discussion and look forward to continued engagement.

---

### Meta-Review · Area_Chair_S55x · 2025-12-23

**Summary:**

This paper introduces a new benchmark for spatial reasoning using multi-view images.  Overall the benchmark appears sound and the authors well articulate what would be gained using their benchmark.  The reviewers provide a strong recommendation for the paper to be accepted, the AC finds any reason to overturn the paper decision insufficient to overrule the reviewers recommendation.

Note that the waymo link provided in the submission is incorrect, please use the correct link in the camera ready:

https://github.com/waymo-research/waymo-open-dataset/blob/master/LICENSE

**Reviewer Concerns:**

The main concern that is still outstanding is the relatively small scale of the dataset.

**Reviewer Scores:**

The authors rebuttal on the data scale is not convincing that it has been resolved, as the authors more or less admit that it is true. Instead, the authors basically say that it is expensive to scale, but this is also the challenge with manual annotations like those this benchmark touts- they're expensive.  As such, it limits the impact of the work as it cannot be easily scaled or applied to other settings, and the data provided has limited uses.

Further- the prices they quote, while could be seen as a substantial investment, but is an expense that is smaller than large swaths of existing benchmarks. As such, the cost quoted is far less than say, the expense of putting together a new object detection or segmentation dataset, as the expectations of scale there are much higher.  This is a substantial problem with the recent trend of putting together small scale datasets on spatial reasoning that may not generalize and have questionable utility as a result.  That said, this is a shared burden of this topic of late, and there is no reason that we should draw the line on this benchmark and not others.  Until the community shows they expect more as a benchmark to be useful I see no reason to reject work like this.

---

### Decision · Program_Chairs · 2026-01-26

Accept (Poster)